# Subsidence more than doubles sea-level rise today along densely populated coasts

**Julius Oelsmann** [1,2] ✉, **Robert J. Nicholls** [3,4], **Daniel Lincke** [5], **Marta Marcos** [6], **Manoochehr Shirzaei** [7,8,9], **Laura Sánchez** [1], **Leonard Ohenhen** [10], **Denise Dettmering** [1], **Jochen Hinkel** [5,11], **Benjamin P. Horton** [12] **& Florian Seitz** [1]

Despite its strong influence on relative sea-level (RSL) rise, there is still low confidence in estimates of vertical land motion (VLM) and its contribution to RSL change. To address this problem, we synergize diverse VLM data, which now cover almost 65% of the coastal population, and are key to resolve small scale subsidence, including East, South, and Southeast Asian cities and populated deltaic regions, largely not covered by earlier geodetic measurements. We find that the average modern (1995-2020) global RSL rise experienced by coastal populations (6 mm/year) is about twice the climate-driven absolute sea-level rise. This reflects a strong tendency for higher rates of subsidence in densely populated areas, with 71% of the global coastal population living in subsiding regions. Paired with community efforts to extend consistent observations, these data are essential to ensure reliable estimates of present and future RSL rise to support risk and adaptation assessment.

With potential damage and protection costs ranging from hundreds to thousands of billions of dollars per year by the end of the century, sea-level change is expected to be one of the most costly consequences of climate change[1]. Global projections commonly focus on scenario-dependent changes and uncertainties of absolute (geocentric) sea-level change, mainly due to thermosteric and mass changes, taken from climate, ocean, glacier, and ice-sheet models (e.g., ref. 2). Coastal impacts and adaptation needs are, however, driven by local (or relative) sea-level rise, i.e., the change of sea level relative to land, which is additionally affected by local coastal vertical land motion (VLM)[3–5]. Observations show that VLM can exceed contemporary absolute sea-level (ASL) changes by as much as an order of magnitude (or more) in susceptible areas, like global deltas and especially coastal cities located on deltas[3,6–9]. However, there is low confidence in state-of-the-art projections of VLM and their uncertainties at the global scale (e.g.,

refs. 2,10). This mainly reflects incomplete understanding and attribution of VLM processes, non-linear changes, the spatial variability, and the often limited availability of observations in time and space, especially in the most densely populated coastal regions[7,11,12]. As many spatial observational gaps are being filled with new data[8,9,13–15], we seek to synergize globally available VLM data to better understand its influence on relative sea-level (RSL) estimations, especially for densely populated coasts where the implications are greatest.

VLM is driven by various natural and anthropogenic processes. Human-induced sub-/surface fluid withdrawal of groundwater, oil, and gas, or drainage has been the major driver of coastal land subsidence in many densely-populated susceptible coastal regions (usually deltas and alluvial plains) over the last century (e.g., refs. 9,11,16–18). Largely groundwater-extraction–related subsidence rates of up to 10 mm/year have contributed to much higher RSL changes than the global average

[1]Deutsches Geodätisches Forschungsinstitut, Technische Universität München, München, Germany. [2]Department of River-Coastal Science and Engineering, Tulane University, New Orleans, LA, USA. [3]Tyndall Centre for Climate Change Research, University of East Anglia, Norwich, UK. [4]School of Engineering, University of Southampton, Southampton, UK. [5]Global Climate Forum, Berlin, Germany. [6]IMEDEA, (UIB-CSIC), Esporles, Spain. [7]Department of Geosciences, Virginia Tech, Blacksburg, VA, USA. [8]Virginia Tech National Security Institute, Virginia Tech, Blacksburg, VA, USA. [9]Institute for Water, Environment and Health, United Nations University, Hamilton, ON, Canada. [10]Department of Earth System Science, University of California, Irvine, Irvine, CA, USA. [11]Resource Economics Group, Albrecht Daniel Thaer-Institute and Berlin Workshop in Institutional Analysis of Social-Ecological Systems (WINS), Humboldt-Universität zu Berlin, Berlin, Germany. [12]School of Energy and Environment, City University of Hong Kong, Hong Kong, SAR, China. ✉e-mail: julius.oelsmann@tum.de

in deltas, such as in the Mekong delta[19–21], or cities built on low-lying deltaic and coastal plains such as Bangkok, Manila, or Jakarta[22–24]. Note that there are a few human-induced uplift processes, with some exceptions (e.g., ref. 25), so the effect of human-induced VLM acts overwhelmingly in one direction—subsidence and hence RSL rise. In addition to anthropogenic processes, natural VLM sources include global Glacial Isostatic Adjustment (GIA) signals, mass loading changes, tectonic processes, volcanism, and mantle dynamics (e.g.,[26–29]), as well as sediment loading and compaction[11,30,31], which are also widely influenced by human activities.

Despite the large influence VLM has on RSL rise, previous global-scale analyses of coastal exposure and damages generally relied on synthesis of the VLM literature (of human-induced subsidence rates, e.g., refs. 19,32), reflecting the limited availability of direct VLM observations. Based on such estimates, Nicholls et al.[33] (hereafter NI21b) highlighted that the coastal population (living below 10 m above sea level) is preferentially concentrated in subsiding areas, especially in susceptible cities and deltas. They estimated that the global coastal population-weighted RSL rise (7.8–9.9 mm/year) strongly exceeds the global coastal average (2.6 mm/year), establishing this study as an important benchmark that demonstrated the global-scale significance of subsidence for coastal residents. However, one of the limitations of NI21b is that no direct geodetic VLM measurements (e.g., by InSAR (Interferometric Synthetic Aperture Radar) or GNSS (Global Navigation Satellite System)) were considered. Instead, subsidence rates were compiled from numerous individual studies and meta-analyses (e.g., ref. 32) that rely on heterogeneous approaches, including borehole extensometers, leveling surveys, groundwater-extraction-based models, etc., or simplified assumptions. Thus, additional limitations of these estimates include their low spatial resolution, particularly in large deltas and cities, where they are often represented by single values averaged over entire regions, thereby neglecting small-scale variability, as well as their frequent reliance on expert judgment, especially in deltas where observational constraints were sparse. Accordingly, the impact of VLM—as determined by currently available high-resolution observing systems—on contemporary RSL change and human exposure to sea-level rise is unclear.

This paper addresses this limitation and provides a global-scale assessment of modern RSL rise, leveraging the multitude of existing and recently published observations and models. Among the globally available observations, GNSS measurements provide the most accurate and precise VLM estimates (e.g., refs. 6,34,35). However, many Asian and African coastal cities and deltas where coastal populations are concentrated are poorly instrumented, or the data is not publicly available (e.g., in China or India, see also Becker et al.[21]). Hence, indirect estimates from the differences between ASL changes and altimetry and RSL changes from tide gauges have also been developed[3,36,37]. While these techniques represent sparse point-wise measurements, InSAR can provide velocity estimates in the satellite's line of sight for a large number of coherent reflectors, such as buildings or infrastructure. Using known ground velocities, typically derived from GNSS measurements, these line-of-sight (LOS) velocities can be transformed into two-dimensional velocity estimates in a geodetic reference frame at spatial resolutions of several meters. Provided that the vertical motion of reflectors (e.g., such as bare rock or pavements) represents the motion of the land surface and that the data are accurately calibrated, these velocities can be interpreted as VLM[38]. Accordingly, InSAR is being increasingly used in many large coastal cities or more widely[13,15,22,39–42], but it remains strongly dependent on the quality and availability of underlying GNSS estimates[6,43].

Although previous studies have synergized some of these data sources, such as GNSS, tide gauges, altimetry data, and InSAR[6–8,44,45], to date, no study has combined all of these observational techniques to assess the influence of VLM on RSL change on a global scale. Several of these currently available observations (in particular InSAR and GNSS) have also not been included in studies focused on sea-level change projections and impacts, as they mainly relied on GIA models or tide gauge VLM reconstructions, which poorly resolve VLM at the regional to local scale, especially in South, Southeast, and East Asia[2,10,46–49]. Therefore, we take a pragmatic hybrid approach and compile the most comprehensive VLM dataset from the available sources, revealing small-scale VLM (i.e., several hundreds of m), including the most populated coastal metropolitan areas, and analyze current changes of RSL rise (see "Methods"). While these observational estimates provide important constraints on present-day RSL and for coastal risk assessments, we recognize that these data sources capture only the surface expression of VLM. Since VLM is driven by a variety of subsurface processes, which are influenced by geological conditions, resource use, and subsurface management, a full understanding and projection of VLM thus requires integration of geodetic observations with subsurface data and process-based models (e.g., refs. 11,50,51).

The aim of the paper is to provide the most robust estimates of contemporary RSL change with the available data. It builds on NI21b using a methodology that integrates the best available VLM data in a hybrid, comprehensive approach. We seek to answer: (1) What is the current (1995–2020) rate of RSL rise experienced by coastal populations? (2) How much of these changes is driven by ASL rise and how much by VLM? and (3) What are the current observational limitations and uncertainties? Based on the results, we provide recommendations for the VLM observational network in order to improve the understanding of RSL rise.

## Results

### Global hybrid vertical land motion estimates

The hybrid VLM estimate is based on four different data sources. We use an interpolated VLM reconstruction based on the joint analysis of GNSS, tide gauges (TGs), and satellite altimetry (from Oelsmann et al.[7], hereafter referred to as OE24) and a GIA model (from Caron et al.[52]). To refine the resolution of VLM in major coastal cities and the largest deltas, where VLM is often largest, we exploit InSAR VLM data from the European Ground Motion Service (EGMS, see also Thiéblemont et al.[13]), from Hamling et al.[53], Ohenhen et al.[8,9], Shirzaei et al.[14], and Ao et al.[15], which resolves subsidence at very small scales, and in the most populated areas where no GNSS (or tide gauge) station data are publicly available. In addition to InSAR, we use GNSS estimates (from the Nevada Geodetic Laboratory, ref. 43) for densely populated areas, where no InSAR data are available (see Figs. 1 and S1; more details can be found in "Methods"). All datasets are aggregated to the 12,148 coastal segments of the Dynamic Interactive Vulnerability Assessment (DIVA) model[54], as used by NI21b.

Here, we rely on several assumptions: First, we assume that all reported InSAR rates reflect VLM. While this is supported by the estimated accuracies of the datasets from validations with GNSS measurements, which are usually between 1 and 2 mm/year (see also "Methods," Fig. S2 and Table S1), several factors may violate this assumption. In particular, there is uncertainty regarding the extent to which InSAR-derived subsidence rates may be underestimated or misinterpreted, either because shallow subsidence is only partially observed (when reflectors such as buildings are founded on deeper layers), or due to the influence of vertical accretion in dynamic, non-urban landscapes[12,38]. Estimates of these components are usually not available on a global scale. One exception is the Mississippi Delta, where a dedicated subsidence map based on RSETs (Rod-surface elevation tables) is available[55] that represents both shallow and deep subsidence, and is thus incorporated for this region in this study. Our second assumption is that trends from all data sources are representative of the changes over the entire period (1995–2020) considered here. This is a simplification, as InSAR-based VLM rates are generally derived from relatively short and variable observation

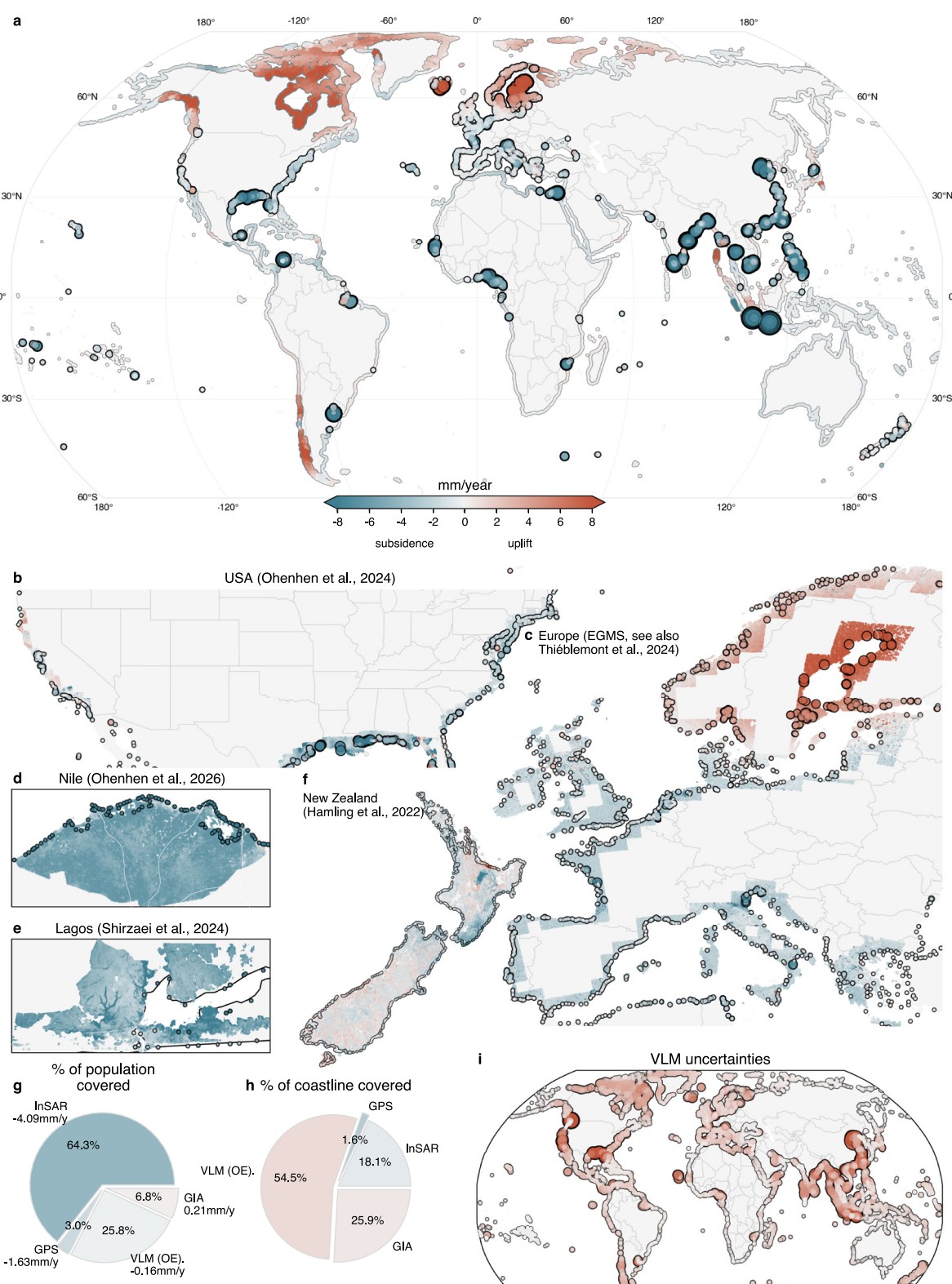

**Fig. 1 | Hybrid estimate of vertical land motion (VLM) along the global coastlines.** The main map (**a**) shows uplift (positive signals) and subsidence in mm/year on the DIVA coastal segments. Values outlined by black circles present regions where InSAR data is available, or where single GNSS estimates are used. The points are scaled by their absolute value to highlight strong subsidence or uplift. Individual InSAR estimates for: **b** the United States[8], **c** Europe from EGMS (see also Thiéblemont et al.[13]), **d** the Nile Delta[9], **e** Lagos[14], and **f** New Zealand[53]. For Europe and the United States, we show the coastal low-resolution DIVA grid points (12,148

elements), whereas for the other regions we show the high-resolution grid (247,666 grid points; see also "Methods"). **g**, **h** The fraction of the coastal population (length) covered by the individual datasets (InSAR, GIA, GPS, and the interpolated data from OE24). The colors depict the population- (or length-) weighted vertical land motion (in mm/year) of the individual datasets. **i** 1$\sigma$ VLM uncertainties of the datasets (using the same color scale as in **a**), which are computed from the formal, spatial, and cross-validation uncertainties in the InSAR data, and the provided uncertainties from OE24 and ref. 52 (see "Methods").

periods (~5–15 years, see SI Table S1). Uncertainties arising from potentially unobserved non-linear changes, as well as from partially observed shallow subsidence and vertical accretion, are therefore discussed further in the Discussion.

Thanks to the abundance of recently published InSAR datasets, almost 65% of the global coastal population is now covered by accessible measurements (see Fig. 1). Regions where InSAR VLM estimates have been processed (highlighted by black-outlined markers) cover almost the entire US coast[8], large parts of Europe (EGMS), many Chinese[15] and other large coastal cities and deltas mainly in South, Southeast and East Asia[9,14]. These regions contain most of the coastal areas with large populations, i.e., the 40 largest deltas/estuaries, and 34 of the 48 largest coastal cities (as discussed by ref. 56, see SI Fig. S2). For the remaining 14 cities and deltas, we either use GNSS rates when available or rely on interpolated results (OE24). The present availability of InSAR data represents a substantial progress and an opportunity to observe highly localized changes, which were not possible with the existing GNSS network alone, or derived products (ref. 6, OE24). The fact that InSAR only covers about 18% of the global coastline by length, but almost 65% of the coastal population, underlines its utility to resolve VLM for human and socio-economic analyses (ref. 8).

The global hybrid VLM estimate demonstrates that coastal regions with higher population densities are, on average, subsiding (Fig. 1). Local subsidence hotspots are East/Southeast Asian cities like Jakarta (−13.7 mm/year), Tianjin (−13.5 mm/year), Bangkok (−8.5 mm/year), or African cities such as Lagos (−6.7 mm/year) and Alexandria (−4 mm/year). These estimates are derived from InSAR observations [see also SI Fig. S3] and represent coastal averages of the aggregated VLM data on the DIVA grid. However, subsidence rates can vary substantially within some of the fastest-subsiding cities. In Jakarta, for example, some areas subsided at rates of up to −42 mm/year, while more central parts experienced uplift of up to +15 mm/year [0.1th and 99.9th percentiles]. Similar, though generally less pronounced, spatial contrasts are also found in other cities such as Bangkok and Ho Chi Minh City. As a result, aggregated coastal city-scale estimates remain associated with considerable uncertainty, and local risk assessments require careful consideration of high-resolution spatial variability, as well as the specific infrastructure and populations affected. Such local effects clearly cannot be resolved with previous interpolated datasets based on GNSS (OE24), nor GIA models, and therefore strongly benefit from InSAR data as local information.

In addition to coastal cities, deltas are especially vulnerable to sea-level rise [refs. 20,30; NI21b; refs. 9,21]. The InSAR estimates[9] and dedicated external data sources of subsidence, including estimates based on RSETs and GNSS, (ref. 55, see also SI Fig. S5 for an overview) currently cover coastlines that contain a population of 389 million people, i.e., almost 90% of the global delta LECZ (according to the DIVA estimates), which presents a substantial improvement in terms of coverage and consistency, especially for the Southeast-Asian deltas[21]. We find that most of the largest deltas are subsiding with average rates (and spatial standard-deviations) of −5.4 (3.6) mm/year in the Ganges/ Brahmaputra delta, −7.8 (2.9) mm/year in the Nile delta, −2.7 (2.7) mm/ year in the Yangtze delta, and −5.6 (5.5) mm/year in the Mekong delta (see SI Fig. S5). While these statistical averages (over the entire deltas) are useful for a general overview of delta subsidence, they do not represent the substantial spatial variability of subsidence within the deltas, which can reach values of up to 12 mm/year (95th percentile) in the Ganges or Nile delta, for instance. Hence, the new InSAR estimates are crucial for assessing these variations, which have so far been hindered by the poor coverage by GNSS stations (SI Fig. S5).

While densely populated coastlines often experience the highest subsidence rates, we also find that these regions are associated with the largest uncertainties. VLM uncertainties can be influenced by non-linearities, technique-dependent noise, differences in observation-window length, cross-validation uncertainties (e.g., potential offsets

between InSAR and GNSS rates), spatial variability and aggregation effects, and parameter uncertainty. Here, we integrate formal, spatial, and cross-validation uncertainties for InSAR VLM data (based on the comparisons with GNSS trends, SI Table S1), as shown in Fig. 1d and explained in more detail in the "Methods," SI Figs. S6d and S7. As a result, median VLM uncertainties are generally higher in the most populated cities (2.6 mm/year) and deltas (1.8 mm/year) compared to all other regions (0.9 mm/year), largely due to InSAR uncertainties (SI Fig. S6d). Locally, VLM uncertainties can reach 7–10 mm/year, particularly in some of the most densely populated cities, such as Jakarta and Tianjin (see SI Fig. S3).

## Implications of VLM-driven relative sea-level change for global coastal populations

To understand how subsidence enhances coastal RSL rise, we combine the hybrid VLM estimates with the ASL change (using gridded altimetry data from the Copernicus Marine Service, see "Methods") and compute the contemporary RSL change rates (Fig. 2a). The spatial patterns of these changes are compared to those of the coastal population living below 10 m above sea level in the LECZ, see Fig. 2b. Most of the densely populated LECZs are situated in East, Southeast and South Asia (from Japan to Pakistan, including Malaysia and Indonesia). In these areas, the ASL change is slightly greater than the global mean sea-level change (not shown), due to redistribution of mass and associated gravitational and rotational effects, and changes in the ocean circulation (e.g., ref. 57). GIA plays a small role in these regions compared to Europe and North America[58].

We follow NI21b and consider the global population-weighted mean estimates of averaged RSL change to understand what the average coastal resident experiences, as opposed to what the average coastal area experiences, which is how sea-level data is normally weighted. This distinction is important because coastal populations are highly unevenly distributed, with large numbers of people concentrated in low-lying urban and deltaic regions where RSL change can differ substantially from the mean (weighted by coastal length). Population weighting, therefore, provides a more appropriate measure for assessing human exposure to RSL change and the associated contribution of subsidence to RSL hazard on a global scale. By contrast, length-weighted estimates may be more relevant for applications focused on coastal land changes. The population-weighting is based on coastal floodplain population estimates from the DIVA model. Fig. 2d and SI Fig. S4 show population-weighted averages of RSL changes for different countries. The highest average population-weighted RSL change affects countries, such as Thailand, Bangladesh, Nigeria, Egypt, China, and Indonesia, at about 7 mm/year to 10 mm/year. The USA, Netherlands, and Italy also experience enhanced population-weighted RSL change of about 4–5 mm/year. In a few nations, geological uplift mitigates some of the current ASL change, such that lower-than-average, or even negative population-weighted RSL change rates occur (e.g., Sweden, Finland). Clearly, these country-averaged values strongly depend on the sub-national distribution of population centers and VLM, and there is often substantial variability within countries. As an example, the population-weighted standard deviation of RSL change can be as large as 7–9 mm/year for countries like China or Indonesia (see SI Fig. S4), and differences between the US Gulf and West coast (as shown in Fig. 1a) are not resolved at the country-aggregation scale. Therefore, the effects of subsidence need to be understood and quantified at a sub-national scale, and this is important to support risk assessment and evidence-based policy making[5,59].

The geographical distributions of RSL trends and population density in Fig. 2a, b also indicate that, globally, higher-than-average RSL changes are generally more frequent in regions with higher population. The cumulative distribution of the population-weighted contribution of VLM to RSL change underlines this disproportionate

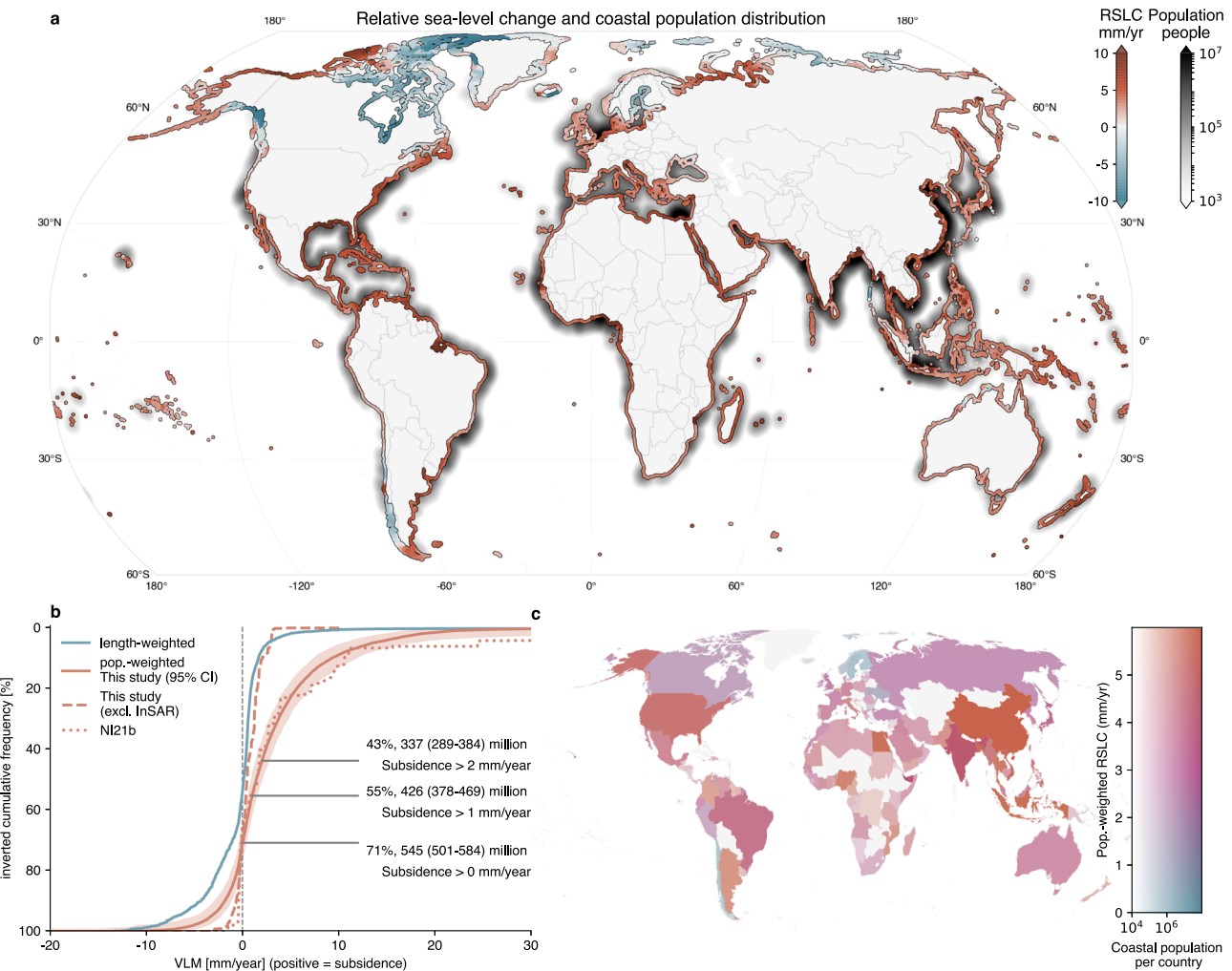

**Fig. 2 | Contribution of VLM to relative sea-level change. a** RSL change [mm/year] as the combination of ASL change (from CMEMS over 1995–2020) and VLM (based on the VLM reconstruction OE24), GIA (Caron et al.[52]), where OE24 has missing data) and InSAR from EGMS[8,9,14,15]. We also show the number of people living below 10 m elevation on a logarithmic scale (black colorbar). For illustrative purposes, we applied radial basis function smoothing with a 120 km length scale to the coastal population data at the DIVA segment grid points to emphasize global population hotspots. **b** Cumulative distribution of the contribution of coastal length-weighted and population-weighted VLM to RSLC (positive sign = subsidence). The inverted cumulative frequency on the *y*-axis thus refers here to the share of the population experiencing at least the subsidence rate shown on the *x*-axis. Solid lines represent the hybrid VLM estimates of this study, the dashed line represents the same dataset

without InSAR, and the dotted line represents the total averaged VLM estimate from NI21b. The shaded region surrounding the solid red line (i.e., the VLM estimate) denotes the 95% confidence interval. This uncertainty is estimated from a bootstrapped distribution of CDFs generated by perturbing the VLM rates using normally distributed random errors derived from the estimated uncertainties (see "Methods"). We also show the fraction of people who experience subsidence rates of >0, >1, and >2 mm/year (using estimates from the hybrid reconstruction). **c** Both the averaged, population-weighted RSL change per country (as indicated by the colors ranging from blue to red, i.e., low to high RSL change), as well as the total coastal population, which is displayed by the modulation in the transparency of the colors.

effect of subsidence on the LECZ population (Fig. 2c): We estimate that 43% of the LECZ population is affected by subsidence of at least 2 mm/year, 55% experience rates of at least 1 mm/year, and 71% are overall subsiding. In contrast, areas that are uplifting by at least 1 mm/year contain less than 10% of the LECZ population, indicating that processes such as GIA, or tectonic uplift, as well as managed groundwater recharge (e.g., ref. 11), currently only attenuate RSL rise for a small fraction of the LECZ population. Earlier geodetic estimates (OE24) that do not include InSAR data substantially underestimate—or simply do not capture the extensive subsidence effects on human populations (see Fig. 2c, dashed line). As an example, the estimated number of people affected by subsidence greater than 2 mm/year is about ten times larger when integrating the new InSAR estimates, compared to using state-of-the-art geodetic data. In contrast, at the upper end of the distribution (for subsidence values greater than 5 mm/year), results by NI21b (dotted line, representing the average between their upper and

lower estimates) substantially overestimate subsidence compared to the hybrid VLM observations (see also discussion). These results reveal that, although subsidence can be a very localized effect, it has global implications, as the majority of the global coastal population is experiencing subsidence.

To consider the global-scale contributions of VLM to RSL change, we compare the coastal-length versus the population-weighted global RSL trends, following NI21b (see Table 1 and Fig. 3). Here, we use different VLM data combinations to analyze the role of different components and approaches on the estimates of RSL changes worldwide. When only using the VLM reconstruction data (from OE24, together with GIA estimates at locations with missing data), the population-weighted average RSL change (3.8 mm/year) is about twice as large as the coastal-length-weighted average (1.9 mm/year), which is mostly caused by VLM, with a minor contribution from ASL change (see second and third rows in Table 1). When including InSAR and GNSS

## Table 1 | VLM, ASL, and RSL change in [mm/year]

|  | ASLC | OE24 + GIA | | OE24 + GIA + GNSS + InSAR | | OE24 + GIA + delta + city (NI21b) | |
|---|---|---|---|---|---|---|---|
|  |  | RSLC | VLM | RSLC | VLM | RSLC | VLM |
| Mean (weighted by coastal length) | 2.76 | 1.88 | −0.88 | 2.09 | −0.67 | 2.03 | −0.73 |
| 1 (weighted by coastal length) | 1.98 | 3.58 | 2.67 | 3.91 | 3.05 | 4.11 | 3.37 |
| Mean (weighted by population) | 3.15 | 3.77 | 0.62 | 5.98 | 2.83 | 8.62 | 5.46 |
| 1 (weighted by population) | 1.10 | 1.31 | 1.18 | 4.18 | 4.06 | 12.25 | 12.32 |

Provided are the mean and standard deviation of coastal-length (coastal l.) and population-weighted data. RSL change and VLM data are given for different VLM dataset combinations. Positive (negative) VLM values indicate subsidence (uplift), or a positive (negative) contribution to RSL change. In the last two columns, we only use delta and city estimates from NI21b, together with the large-scale VLM information from GIA and OE24.

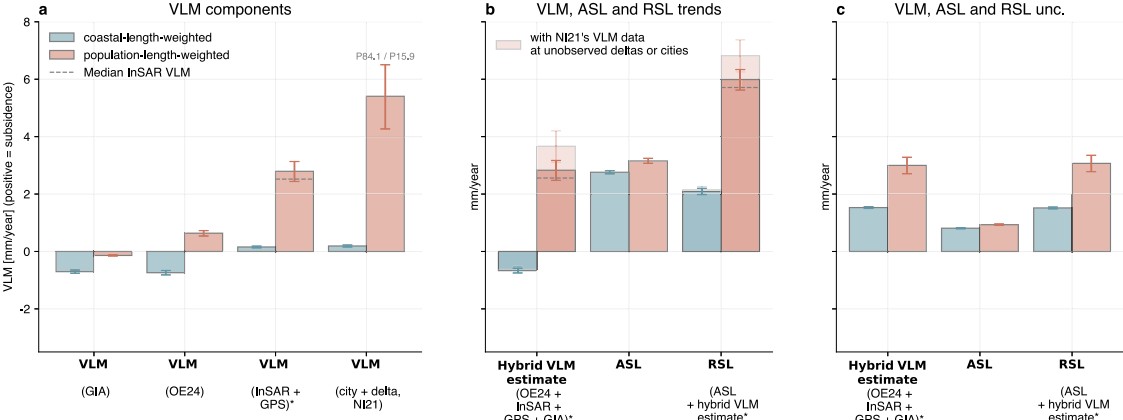

**Fig. 3 | Global average VLM and ASL contributions to RSL change. a** Global weighted averages of different VLM estimates (GIA (Caron et al. 2018[52]), an interpolated estimate (OE24), InSAR and GNSS (only), and the city and delta estimates from NI21b using different weightings. The data is shown in terms of its contribution to RSL change (i.e., positive sign for subsidence, and vice versa). Here, we assume zero VLM at every coastal segment where an individual VLM dataset does not provide any data. **b**, **c** The global weighted-average trends and 1$\sigma$-uncertainties of different dataset combinations (see also "Methods"). The hybrid VLM estimate combines the components (GIA, OE24, InSAR, and GNSS) shown in (**a**). The transparent bars represent weighted averages for the hybrid VLM estimate, where each city and delta VLM data point is replaced by the averaged NI21b results if not directly observed by InSAR. The dashed line shows the global weighted averages obtained when using median rather than mean VLM rates during the aggregation of the InSAR data from the high-resolution to the low-resolution grid. The error bars represent the uncertainties of the global weighted averages as 1$\sigma$, corresponding to the 15.9th and 84.1st percentiles, of a distribution obtained through bootstrapping. This distribution was generated by repeatedly computing the weighted averages of random samples, each containing 50% of the original data, over 10,000 iterations. All datasets marked with an asterisk (*) include random perturbations based on their respective uncertainties, in addition to the resampling (see "Methods," or SI Fig S7). These perturbations are derived directly from the VLM uncertainties; for RSL change, they are based on the square root of the summed squares of the VLM and ASL-change uncertainties.

data estimates in large cities and deltas, this discrepancy increases substantially, such that the population-weighted RSL change of 6 (5.6–6.3) mm/year becomes about three times as large as the coastal-length weighted RSL change (2.1 (2.0–2.2) mm/year). The 1$\sigma$ uncertainties of these averages are derived from a bootstrap analysis that accounts for data heterogeneity and individual data uncertainties (see "Methods"). Therefore, subsidence (with a global population-weighted average of 2.8 mm/year) currently contributes almost as much to the population-weighted RSL change as the climate-driven ASL component (3.15 mm/year). In the future, the accelerating climate-driven ASL change is widely expected to become increasingly dominant (e.g., refs. 2,60), but future subsidence rates are uncertain and need more assessment. Human-induced subsidence could increase, but equally could be reduced by active subsidence control.

Uncertainties in the global population-weighted mean subsidence (Fig. 2b) are influenced by the estimated VLM data uncertainties (contributing to ±0.45 mm/year, Fig. 2c) and by the spatial heterogeneity of VLM and population (contributing to ±0.32 mm/year). The choice of the averaging method also affects the global statistics: Using median InSAR rates instead of means when aggregating the data from the high- to the low-resolution DIVA grid (see "Methods") reduces the population-weighted mean VLM by 0.27 mm/year (see also SI Fig. S7

for an overview). Since the population-weighted average VLM uncertainties of 3 mm/year (1$\sigma$) are substantially larger than the ASL trend uncertainties (0.9 mm/year, Fig. 2c), they dominate uncertainties in RSL estimates. However, the fundamentally different approaches to defining uncertainty in the VLM still hinder an objective comparison and therefore require consistent frameworks to assess these differences.

Besides these uncertainties, regional variations in the rates are strongly enhanced when comparing the coastal-length (population) weighted standard deviations of VLM, i.e., 3.1 mm/year (4.1 mm/year), with the standard deviations of the ASL change, i.e., 2.0 mm/year (1.1 mm/year). This is partially caused by single extremes in regional or local subsidence or uplift, leading to much longer tails in the distribution of RSL change. Therefore, VLM is the dominant driver of regional variability of current RSL changes (see also Table 1).

When we replace the InSAR and GNSS data in these estimates (OE24 + InSAR + GNSS + GIA) with the literature-based delta and city subsidence data from NI21b (i.e., the average of the upper and lower estimates), we find an even higher population-weighted RSL change (8.6 mm/year, see Table 1). This is mainly because the subsidence rates are higher in the combined literature-based city and delta data from NI21b than in the InSAR data (particularly at the upper end of the

distribution, see Fig. 2c). Here, it should be noted that NI21b considers more city and delta VLM data compared to the InSAR data (which is provided at some of the largest coastal cities and deltas, but not all). Hence, to be consistent, we also compared the RSL and the hybrid VLM estimates only at the locations where both datasets contain estimates. Since this comparison yields similar results, it confirms that the city subsidence estimates from NI21b on average exceed the VLM estimates from InSAR observations. This strongly suggests that the literature-based VLM estimates are biased towards high values and stresses the benefits of a sample of VLM from InSAR measurements.

Although there are large differences between these datasets, these results reinforce the important insight that RSL rise often increases with coastal population density, mainly due to the contribution of subsidence. However, it should be emphasized that this relationship is strongly skewed, and large coastal cities (e.g., Tianjin, Jakarta, or Bangkok) and densely populated deltas (e.g., Nile and Ganges–Brahmaputra) are affected by strong subsidence and thus increased RSL change. This finding is also supported by Fig. 3a, which shows global averages of the individual VLM contributions to RSL change based on the different datasets (GIA, OE24, InSAR + GNSS, and city + delta VLM from NI21b). Here, we set the coastal VLM data to zero where no data is provided for each individual combination. As can be seen, high subsidence rates based on the global population-weighted averages from InSAR, or city and delta (from NI21b) data strongly contribute to the global population-weighted RSL changes, even though these estimates are only available for a small fraction of the world's coastlines. On the contrary, this effect is much less pronounced when InSAR data is not considered (Fig. 3a) and only an interpolated dataset is used (OE24). This reflects the power-law distribution of coastal population, i.e., about 90% of the global coastal population (in the most densely populated regions) lives in an area that extends along only about 10% of the global coastline (e.g., refs. 61,62). The data indicate that these highly populated regions are statistically more likely to coincide with higher-than-average subsidence on a global scale, which explains why humans experience RSL changes that are significantly higher than the spatially averaged rates. However, understanding the causes of these statistical relationships requires consideration of interacting physical processes and human management decisions, including drainage policy, construction practices, soil settlement, resource extraction, and broader subsurface-use governance.

### Toward community-driven advances in VLM observations, process understanding, and projections

In this paper, we assess the exposure of coastal populations to contemporary RSL changes using VLM observations. Our results qualitatively confirm recent findings of NI21b that, on average, subsidence significantly increases the RSL change experienced in highly populated areas. The contemporary population-density weighted global-mean RSL change of 6 mm/year is nearly three times the coastal-length weighted global-mean RSL change (2.1 mm/year). Since large cities and deltas are the primary epicenters of subsidence, but are usually not well covered by GNSS stations or tide gauges (see also SI Fig. S6a, b), these effects are almost completely missed or underestimated by previous assessments (ref. 6, OE24, see also Fig. 2c) or current IPCC global scale projections (e.g., ref. 2). Therefore, our results confirm that systematic consideration of coastal subsidence is essential to understand global coastal exposure, risks, and costs due to sea-level rise.

Although our analyses qualitatively align with NI21b results, quantitatively, the relative increase between the population-density and the coastal-length-weighted RSL rise is not as large as previously reported by NI21b. This indicates remaining limitations, which can be due to inconsistencies and uncertainties in current VLM observations, the partially poor data-availability, limited understanding of the

underlying processes, and non-linear VLM. Differences with respect to NI21b could be either caused by overestimated subsidence rates in cities and deltas by NI21b, which were mostly derived from the literature and expert judgment rather than geodetic measurements, or by missing observations in the applied VLM observation data due to their partially sparse spatial distribution. NI21b noted the difficulty of extracting average values of subsidence from the literature, as few studies systematically derived this statistic. It is also important to recognize that our analysis here is partial in the sense that InSAR or GNSS measurements are not available at every large coastal city or delta. In contrast, NI21b considered 138 large coastal cities (more than one million people in 2005) and 113 deltas worldwide. However, they relied on expert judgments in 77 deltas and 8 cities, and lacked information on most Chinese coastal cities, which are available to this analysis. When we use NI21b delta and city VLM data wherever no InSAR data is available, we observe an additional increase in the population-weighted RSL from 6.0 to 6.9 mm/year (see Fig. 3b). Hence, given these discrepancies and remaining data gaps, there is an important need for comprehensive global analyses to refine our estimates. These should start with the regions where subsidence has the greatest impact—most notably South, Southeast, and East Asia.

With half of the world's LECZ population living in deltas, recent advances in space-based geodetic monitoring (e.g., ref. 9) represent important milestones in covering these vulnerable landscapes with unprecedented resolution. However, delta subsidence estimates, or InSAR VLM estimates in general, are still subject to several sources of uncertainty. Next to the discussed formal, spatial, and cross-validation uncertainties, which are here treated as random non-biased effects, it remains uncertain whether InSAR measurements fully capture the combined effects of shallow and deep subsidence, or to what extent they are influenced by vertical accretion processes (e.g., ref. 12). These factors can be differentiated using in situ data (RSET, marker horizon records, e.g., refs. 55,63, or vertical well extensometers), which has been done for the Mississippi Delta and elsewhere. However, such local information is not yet available at the scale resolved by the InSAR data, which may also contribute to differences with respect to NI21b, who also aggregated data based on such in situ information. If reflectors such as buildings have different foundation depths, this can lead to high spatial variability due to differential settlement between these reflectors. Buildings with shallow foundations respond mainly to near-surface compaction, while pile-supported structures are coupled to deeper, more stable layers and primarily experience deformation occurring at depth. Because InSAR measures displacement of surface scatterers on buildings, the signal may exaggerate risk for structures founded shallowly while underrepresenting stresses affecting deep-founded buildings. This depth-dependent decoupling introduces uncertainty in hazard interpretation, particularly in cities with mixed foundation systems, and highlights the need to interpret InSAR observations together with subsurface stratigraphy and foundation information. Incomplete observation of shallow subsidence could cause an overall underestimation of the population-weighted average subsidence, since it strongly depends on InSAR estimates from the most densely populated areas and may also contribute to differences with respect to NI21b[38]. Expectedly, we find that the global population-weighted average subsidence (as provided in Table 1) is most sensitive to possible biases in the city and delta datasets: A hypothetical bias of 1 mm/year in either of these datasets would translate to a ~ 0.3 mm/year bias in the global population-weighted averaged VLM (SI Fig. S7).

Such remaining uncertainties underline the importance of dense, continuous global GNSS networks (e.g., as provided by the Nevada Geodetic Laboratory, ref. 43). These measurements are the prerequisite to derive comparable InSAR VLM estimates in a global reference frame and are thus fundamental to reducing biases in regional estimates. Yet many of the world's most populated cities and

deltas—particularly in South, Southeast, and East Asia, as well as major African coastal centers—still lack the observational coverage available in Europe or the United States (e.g., ref. 9; SI Fig. S6a, b), further emphasizing the need for continuous and global observations[43]. Comparable to limitations in InSAR estimates, GNSS reference stations with deep anchor depths may overlook shallow compaction in landscapes where such processes are significant, leading to underestimated total subsidence. Another limitation is that GNSS stations only provide point measurements and, unlike InSAR, cannot resolve fine-scale deformation patterns. Therefore, given the key role of deltas and cities in controlling the rates of sea-level rise felt by the average person, these factors must be considered, and InSAR data should ideally be analyzed in synergy with in situ (e.g., extensometers and GNSS) observations.

While these observational limitations are a major challenge in predicting future changes, in many cases, the non-linear nature of the processes themselves can affect the extent to which these changes can be projected into the future[7,11,22]. Especially when subsidence is human-induced, rates can be highly non-linear, as is the case for some observations in Tianjin (SI of Ao et al.[15]), Manila[22], or other large coastal cities and deltas[9,20,23]. Tectonic effects have a significant influence on many coastlines, in particular around the Pacific Rim, and many locations in Southeast Asia and the Mediterranean, but are largely unconsidered in most global studies focused on future sea-level change[2,10,64]. VLM from tectonics varies between co-seismic, post-seismic, inter-seismic, and pre-seismic stages of the earthquake deformation cycle. The spatial footprint of earthquakes can be as large as hundreds to thousands of km, and abrupt co-seismic displacements in major subduction zones can be in the order of meters, as was the case for the 2010 Maule Earthquake[65] and the 2011 Tohoku earthquake (e.g., ref. 66). A recent study demonstrated that earthquake-induced subsidence, when combined with climate-driven sea-level rise, triples coastal flood exposure along the Cascadia subduction zone of the Pacific Northwest by 2100[67]. Ideally, the vertical changes caused by these different stages should therefore be adequately parameterized and interpreted alongside tectonic models and geological records[41,68–70]. However, such non-linear changes are poorly quantified in most of the incorporated datasets. This is partly due to the simplified models used to estimate the VLM, the sometimes short duration of observations (especially of InSAR data), and, most importantly, the lack of understanding of the processes driving the local VLM.

Thus, in addition to accurate and dense VLM observations, the most appropriate and meaningful ways to make projections must be considered (e.g., process models, statistical approaches, and assumptions of future scenarios, such as of groundwater extraction, or sediment supply in deltas[9,31,51,71,72]). Most importantly, the identification of human-induced VLM can motivate measures to mitigate subsidence, e.g., by groundwater regulation (NI21b; ref. 9) at a regional level.

Remaining limitations in current VLM estimates emphasize the importance of open data and community efforts, such as the International Panel on Land Subsidence [IPLS, ref. 73], to improve our process understanding as well as the quality, coverage, transparency, and consistency of the observational database. There is great potential from such efforts to enhance communication within and across the disciplines (i.e., the "sea-level community," solid earth, ocean, social, and economic sciences) necessary to achieve these goals. While recent publications of InSAR observations[8,9,13–15] represent important developments to fill observational gaps along the world's coastlines and to increase confidence in the coastal VLM at a high spatial resolution, community-based efforts could lead to the creation of an open-source platform to provide a systematic overview of, or even combine, individual VLM data sources. Such a platform would represent an essential source of information for (non-)expert users, would contribute to evaluating the quality and consistency of different datasets, and would synergize information that goes beyond the analyses and datasets presented here (e.g., include geodynamic models, geological records, and in situ measurements, such as leveling, etc.).

Here, we show that subsidence causes about half of the RSL change that is currently experienced by coastal populations. Our findings reinforce earlier work highlighting the large potential to reduce RSL change by mitigating human-induced subsidence. We advocate for a reassessment of sea-level projections and impact assessment studies using a hybrid VLM data approach as presented here, incorporating and integrating all available data from different sources. Focusing future improvements on the most heavily populated and rapidly subsiding coastal regions is essential, also because VLM estimates in these areas remain the most consequential for coastal populations and yet are among the most uncertain. This would benefit from joint community efforts, which require the incorporation of the effects of local VLM and realistic and consistent data uncertainty estimates.

## Methods

### Hybrid approach for vertical land motion estimation
We use a combination of observational data from GNSS, tide gauges, satellite altimetry, InSAR, and a GIA model to determine the components of RSL change (see also SI Fig. S1). All datasets (GNSS, altimetry, and InSAR) are referenced to the ITRF2014[74], except for EGMS, which is transformed into ITRF2014 using an empirical transfer function.

### VLM reconstruction
We use the time- and space-resolving VLM reconstruction by Oelsmann et al.[7] as the main information for VLM. The 0.25°-resolution VLM reconstruction is based on more than 10,000 point-estimates from GNSS (in IGS14, ref. 43), TGs[75] and gridded altimetry (as described below) and resolves height changes at an annual time scale over 1995–2020. This probabilistic estimate (hereafter referred to as OE24) was derived in a Bayesian framework using several processing steps. First, the underlying data (GNSS, differences between TG and altimetry) were adjusted/corrected for offsets, single point outliers, and the annual cycle in a semi-automated manner[7,76]. Second, estimates of long-term linear trends and common modes of variability were obtained using a Bayesian principal component analysis[77]. Finally, the spatial components (linear trends and weighting pattern) were interpolated using an adaptive Bayesian transdimensional regression approach[45]. Thus, the resolution of the resulting interpolated maps reflects the underlying spatial distribution of the data. That is, the interpolation has a higher resolution in regions with high station density (e.g., Europe, USA, or Japan), compared to regions with low density (e.g., Africa, India, or China). The final VLM reconstruction was obtained from the sum of the recombined principal components, the interpolated spatial weighting patterns, and the linear trend estimates.

### InSAR and additional in situ data
To obtain the highest possible resolution of VLM, we complement the GNSS + TG + Altimetry-based VLM reconstruction from OE24 with InSAR VLM data for coastal areas in Europe (EGMS) (see also [13]), the USA[8], New Zealand[53], 26 Chinese coastal cities[15], 15 other selected coastal cities around the world[14], and 40 deltas[9]. In the supporting information (SI Fig. S2), we present an additional validation of the datasets, where the VLM data is compared with vertical velocities from GNSS (NGL) by computing an inverse-uncertainty-weighted average of the InSAR data within a 5 km range to the GNSS station. The main information on these datasets is summarized in Table S1.

**EGMS.** To obtain coastal InSAR-based VLM information for most of Europe, we select the coastal tiles of the EGMS Ortho (Level 3) product. Detailed information on the processing can be found in the Algorithm Theoretical Basis Document (see EGMS documentation). The EGMS VLM data were referenced to the European GNSS stations in the

European Terrestrial Reference Frame ETRF2000. To reference these velocities to the International Terrestrial Reference Frame (ITRF2014, ref. 74) and to ensure a consistent reference frame for the entire hybrid dataset, we apply a latitude ($\phi$-dependent transfer function presented by Thiéblemont et al.[13]): $\Delta V_{\text{fit}} = -2 \times 10^{-4}\phi^2 + 0.04\phi - 0.85$. This second-order polynomial transfer function (see also SI Fig. S2) is based on the differences between VLM from GNSS in the ETRF2000 and ITRF2014 reference frames. After applying this transformation to the InSAR data, they are highly consistent with GNSS rates, yielding a weighted standard deviation of 1.19 mm/year based on 1617 stations (see SI Fig. S2).

**US coast and coastal cities.** InSAR data for the US coast and other cities are provided by Ohenhen et al.[8] and Shirzaei et al. (2024)[14]. Both datasets were produced using the same post-processing softwares, i.e., the wavelet-based InSAR (WabInSAR) algorithm[78,79]. To obtain the vertical velocities of the US-dataset, GNSS velocities were interpolated on the pixels, and the LOS 3D displacements were computed by minimizing the differences of the transformed velocities with respect to observed GNSS velocities. To obtain the VLM for the largest coastal cities outside the US[14], the LOS velocities were first transformed into the vertical direction using the incidence angle. Next, to transform the data into a global reference frame (i.e., the ITRF), the InSAR-based VLM was constrained by the global GNSS-based VLM map of Hammond et al. (2021)[6], which is based on the ITRF2014. Both datasets were validated with respect to VLM from GNSS stations [NGL, ref. 43]. Comparing the US-data with 756 GNSS stations, Ohenhen et al.[8] reported a standard deviation of the differences of 1.5 mm/year. Likewise, the comparison of rates at 17 coastal cities resulted in discrepancies within the range of ±1 mm/year[14]. These reported accuracy levels are consistent with additional validations performed within this study (see SI Fig. S2), which yield weighted standard deviations of 1.43 and 1.72 mm/year, for the US ($n = 943$) and city ($n = 115$) datasets.

**New Zealand**

Hamling et al.[53] derived InSAR VLM data at unprecedented resolution for the North and South Islands of New Zealand from Envisat, covering a period from 2003 to 2011. VLM rates were computed by aligning the vertical and horizontal velocities with GNSS measurements over several processing steps that account for the complex tectonic and volcanic settings and velocities (discontinuities, non-linear deformations) in New Zealand (please refer to ref. 53 for more details). Deviations of the resulting deformation rates from GNSS rates have a standard deviation of 1.6 mm/year, comparable to the accuracy of InSAR data for the US and large coastal cities. The VLM trends and formal errors are provided on a coastal profile with a ~1 km resolution, representing spatial averages of data with 5-40 km of the coast.

**Global deltas.** We additionally use InSAR VLM data for 40 of the largest coastal deltas from Ohenhen et al.[9], derived from observations over 2012–2023. These data were produced using the same software as used for the US coast[8]. The LOS velocities were transformed into vertical velocities using GNSS stations as geodetic ties in 17 deltas and coastal plains (i.e., for the Fraser, Mississippi, Rio Grande, Rhine, Rhone, Po, Vistula, Red, Amazon, Parana, Ciliwung, Brantas, Ganges–Brahmaputra, Chao Phraya, Mekong, Pearl, and Chikumagawa delta). For deltas without GNSS stations, they incorporated the interpolated data of Hammond et al.[6]. The validation against rates from 81 GNSS stations revealed a root mean square error of 1.2 mm/year, when the same observation period was used.

For the Mississippi Delta, we use the InSAR estimates (from ref. 8) in the Greater New Orleans metropolitan area and implement a more dedicated VLM estimate in the remaining Delta area. We use an interpolated dataset from Nienhuis et al. (2017)[55], which represents the combined effects of deep subsidence and shallow subsidence in the

delta (excluding vertical accretion). This dataset is based on in situ measurements from rod surface-elevation-marker horizon records that enable estimates of shallow compaction and vertical accretion, as well as GNSS data, that are used to determine deep subsidence. This approach represents the most rigorous method to determine subsidence in deltas; however, similar dedicated products are not available in other megadeltas.

More information on the estimates in deltas can be found in SI Fig. S5, where we show information for all deltas (according to the list in ref. 32), where any geodetic information is available. Here, we report delta averages from InSAR (averaged over the entire delta surface, or averaged over the InSAR data points that are interpolated onto the DIVA segments), as well as from GNSS point measurements, and a combination of InSAR and GNSS. Note that these estimates sometimes differ substantially, which is mainly due to the large spatial variability of VLM, and the heterogeneous coverage of GNSS stations in the deltas. Accordingly, these results cannot be directly compared as they do not represent co-located (InSAR-GNSS) estimates. GNSS estimates for deltas where no InSAR data are available are provided at https://doi.org/10.5281/zenodo.19830369.

**China.** We use the city median and percentile information of VLM that is provided for a set of Chinese coastal cities[15]. For six cities, GNSS observations were used to provide a reference VLM. For the remaining cities, an iterative approach was performed to obtain a reference VLM that approximates the mode velocity of all pixels. The calculated LOS displacements were then transformed into a vertical velocity, taking into account the Radar's incidence angle. A validation of InSAR VLM estimates by Ao et al.[15] with 32 GNSS time series (of the CMONOC network [National Earthquake Data Center, ref. 80]) provides a root-mean-square error of 1.5 mm/year, indicating a similar level of accuracy as for the other datasets considered. Note that we do not provide an additional validation here, as very few stations are publicly available in these Chinese cities. Here, we use the reported RMSE of 1.5 mm/year as a measure of cross-validation uncertainty for all cities. To quantify the uncertainty arising from the spatial variability of VLM rates, we make use of the available percentile information, specifically the 5th, 50th, and 95th percentiles, as well as the percentiles corresponding to rates below 3 and 10 mm/year reported for each city in the Supplementary Information of Ao et al.[15]. Using these percentile constraints, we fit a Johnson's $S_U$ distribution, capable of representing arbitrary skewness and kurtosis. This enables us to approximate the underlying distribution of VLM rates for each city in a way that accommodates both asymmetry and heavy-tailed behavior.

**GIA**

Because the VLM reconstruction has missing values, particularly in northerly regions due to inadequate instrumentation, we fill these gaps using the GIA trend and ($1\sigma$) uncertainty estimates from Caron et al.[52]. This model was derived from Bayesian inferences of the probability distributions of model parameters describing the rheological structure of the Earth and ice history. To generate a likelihood probability distribution of the model parameters, Caron et al.[52] generated an ensemble of 127,000 forward models (with different parameters) and estimated the probability of how likely a given model is to explain the observational constraints. The VLM estimates correspond to the expectation and standard deviation of these probability distributions.

**Combination**

We combine the linear trend estimates of the three data sources (OE24 + InSAR + GNSS + GIA) and map them onto the 12,148 coastal segments of the DIVA model[54], as used by Nicholls et al.[33] (see SI Fig. S1). The different datasets are successively mapped onto the segments using nearest neighbor interpolation (with a maximum allowed distance of 50 km) in four different steps. First, we use all the VLM

trends and (1$\sigma$) formal uncertainties of all InSAR datasets (China[15], coastal cities, New Zealand, USA, Europe, Deltas) and interpolate them with nearest neighbor interpolation on the high-resolution grid of DIVA (consisting of 247,666 segments), wherever available. These segments are then averaged to the lower-resolution grid (12,148 segments) by computing both mean and median values within each cell. In that process, we also compute the standard deviation of all averaged VLM trends, to estimate spatial uncertainty $\sigma_{spatial}$. Next, we incorporate GNSS trends from NGL, ref. 43) at remaining large coastal cities or deltas that are not covered by InSAR and where the population living below 10 m, as provided by the DIVA model, is more than 50 thousand people, as well as for remote islands, where OE24 is not defined or not sufficiently resolved. For the remaining coastal segments, we use trend and 1$\sigma$ uncertainty estimates from VLM reconstruction, as well as from a GIA model[52] wherever the former is not defined. Note that in the following analyses, some of the regions north of Greenland and Antarctica are masked out because they are either not included in the DIVA model or due to missing altimetry data.

## VLM uncertainty quantification

As discussed in the main text, uncertainties arise from various causes: These can be broadly categorized into *random errors* and *cross-validation* uncertainties in InSAR and GNSS observations (e.g., temporal variability in local conditions, or random errors in geophysical corrections), which are ideally reflected in the provided formal trend uncertainties and by the differences between the datasets; *systematic uncertainty* (biases, reference frame offsets between InSAR and GNSS), *methodological* (differences between processing approaches and physical models, e.g., GIA models or the VLM interpolation), *temporal and spatial* uncertainty (different window lengths, spatial variability and aggregation approach), as well as *model-input* or *parameter* uncertainty (e.g., GIA model Earth parameters). Thus, to understand the influence of these factors on our global averaged statistics (specifically the population-weighted global VLM), we compute the following uncertainty metric for all InSAR datasets:

$$\sigma = \sqrt{\sigma_{\text{time, random}}^2 + \sigma_{\text{cross}-\text{validation}}^2 + \sigma_{\text{spatial}}^2} \qquad (1)$$

Here, $\sigma_{\text{time, random}}$ are assumed to be represented by the provided formal uncertainties of the different products, as both window-length and noise amplitude influence formal uncertainties. Note that the reported formal uncertainties may not be fully comparable across datasets, as they are derived using different methodologies. However, the combined uncertainties are usually dominated by the cross-validation and spatial components. Next, we use the standard deviations of the differences w.r.t. GNSS as cross-validation uncertainties $\sigma_{\text{cross}-\text{validation}}$ for each dataset, which are usually within the 1–2 mm/year ranges. Finally, because the data is aggregated from a high (-250k) to a low (-12k) resolution grid, we use the standard deviation of trends at locations that are aggregated to single points on the low resolution grid as a measure of sampling/spatial uncertainty $\sigma_{\text{spatial}}$. Thus, we assume that the data follow a normal distribution. While this is a simplification, it enables a more straightforward combination with the other uncertainty components and avoids the need to estimate multiple distribution parameters for each location. For consistency, we thus estimate the spatial standard deviations (uncertainties) for the Chinese cities based on the interquartile ranges (divided by -1.35) from the derived non-normal Johnsons $S_U$ distributions. Note that since we are not matching every InSAR pixel with high-resolution population information, this spatial uncertainty should also express the uncertainty about the high-resolution distribution of people that are also aggregated on the lower resolution grid.

Uncertainties at all other locations where no InSAR is available are based on the probabilistic trend uncertainties of the interpolated data [OE24], which combines temporal, spatial, and technique-dependent

noise, formal GNSS uncertainties[43], as well as the model uncertainties that are based on a large GIA model ensemble from ref. 52, and thus accounts for Earth-parameter uncertainty. It is important to emphasize that this remains a simplified way of combining uncertainties derived from fundamentally different methods. A more rigorous treatment would require holistic probabilistic frameworks capable of jointly modeling technique-dependent noise and biases, while also resolving the temporal evolution of height changes, which is challenging, especially for the very large InSAR datasets. Such an approach could help to mitigate inconsistencies arising from differing observation windows.

To compute the uncertainty in the population-weighted mean statistics, we also account for uncertainties associated with the spatial distribution or sampling of data using bootstrapping (Fig. 3 and SI Fig. S7). We compute the population-weighted mean for 10,000 randomly selected subsets of VLM estimates, accounting for 50% of the total sample size (i.e., the size of the low-resolution grid 12,148). This uncertainty expresses the sensitivity of the global mean to the heterogeneity of the distribution of VLM rates and population sizes and explores if the mean is also significant for subsets of the data. This uncertainty is larger when the global average statistic is controlled by only a few locations, and vice versa. For each resampled subset, we also perturb the VLM trends by randomly distributed errors drawn from normal distributions, which are scaled by the derived uncertainty metrics (as the standard deviations of the distributions) of the techniques, as explained before. The influence of the resampling, the random perturbations, the different InSAR averaging approaches (mean vs. median), and hypothetical biases (representing systematic uncertainties) in each InSAR dataset are shown in the SI Figs. S6d and S7. It should be emphasized that this analysis isolates only contributions of observational VLM uncertainties from other uncertainty sources present in digital elevation models, protection estimates, and population data, which are fundamental for coastal risk assessments[81,82].

## Absolute sea-level change data

To obtain RSL change estimates, the VLM data are combined with trends from gridded 0.25° satellite altimetry data from the Copernicus Marine Service (CMEMS, https://data.marine.copernicus.eu/product/SEALEVEL_GLO_PHY_L4_MY_008_047/description, last access 24.03.2022). The monthly altimetry ASL anomalies are used from 1995-01-01 to 2020-12-31. Note that these measurements start earlier than most of the InSAR observations. Therefore, the overall global ASL trends over this period are slightly lower than rates over more recent periods, e.g., from 2010 onwards, due to accelerations in the global mean sea level[60]. However, since ASL trends computed over shorter periods (e.g., from 2010) are much more strongly dominated by interannual climate variability (see SI Fig. S8), leading to much lower rates particularly in Southeast Asia, we still use the trends over the entire period. We use a model consisting of trend, offset, annual, and semiannual cycles, to compute the linear rates of ASL changes using linear least squares. As for the VLM data, the altimetry data are mapped onto the coastal segments of the DIVA model (SI Fig. S1).

## Coastal population data

To quantify the exposure of the coastal population to contemporary RSL change, we use population data as applied in Nicholls et al.[33]. Here, we consider the population living below 10 m elevation in 2015, which is 768 million people in total. The digital elevation model data are obtained from Shuttle Radar Topography Mission elevation data[83,84]. It is combined with Global Rural–Urban Mapping Project population data using resampling methods[85,86], and provided at the same segments of the DIVA model.

## Data availability

The global VLM reconstruction from Oelsmann et al.[7] [OE24] is available at https://zenodo.org/records/8308347. GNSS VLM data (Blewitt

et al.[43]) are available at https://geodesy.unr.edu/velocities/midas.IGS14.txt. InSAR VLM estimates for Europe can be downloaded from the EGMS data explorer https://egms.land.copernicus.eu/. InSAR VLM data for the US[8] is provided for different regions: The Pacific coast (available through the Virginia Tech Data Repository at https://doi.org/10.7294/17711000), the Atlantic coast (https://doi.org/10.7294/19350959), and the Gulf coast (https://doi.org/10.7294/22731326). The InSAR city subsidence data from Shirzaei et al.[8] is available at: https://data.lib.vt.edu/articles/dataset/InSAR-Based_Coastal_Land_Subsidence/25864435/1. Delta subsidence data from Nienhuis et al.[55] is available at https://osf.io/m83z4/files/osfstorage. The delta-subsidence data from Ohenhen et al.[9] is available at Zenodo (https://doi.org/10.5281/zenodo.15015923). Subsidence data for Chinese cities (Ao et al.[15]) can be obtained from https://www.science.org/doi/10.1126/science.adl4366#supplementary-materials. The GIA estimates contained in the VLM data are available at https://vesl.jpl.nasa.gov/solid-earth/gia/. The ASLC data were obtained from https://data.marine.copernicus.eu/product/SEALEVEL_GLO_PHY_L4_MY_008_047/description. The information of the coastal segments of the DIVA model (location, population, length) and the VLM estimates of NI21b can be obtained from the source files provided at https://www.nature.com/articles/s41558-021-00993-z#Sec16. Figures are made with data from Natural Earth (free vector and raster map data @ naturalearthdata.com). Different delta and city subsidence, as well as the global VLM estimates of this study, are available at https://doi.org/10.5281/zenodo.19830369.

## Code availability

The code used to process data and generate all figures is available on GitHub at https://github.com/oelsmann/global_hybrid_vlm_estimates and archived on Zenodo (https://zenodo.org/records/19831243).

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

## Acknowledgements

J.O. received funding from the European Union's Horizon Europe research and innovation program under the Marie Skłodowska-Curie grant agreement No 101210999. BH was supported by City University of Hong Kong for the project "No. 36.9. The Asia Sea-Level Knowledge Hub (ASK-Hub)," officially endorsed by the United Nations Educational, Scientific and Cultural Organization (UNESCO) under the UN Decade of Ocean Science for Sustainable Development 2021-2030. M.M. acknowledges the DETECT project (reference PID2021-124085OB-I00 MCIN/AEI/10.13039/501100011033/FEDER, UE). The authors thank Philip Minderhoud, Kay Koster, and two anonymous reviewers for their helpful comments. The authors also thank Marcello Passaro and Sönke Dangendorf for their support. Marcello Passaro greatly supported J.O. throughout his PhD and facilitated many collaborations that were fundamental to this research. Finally, the authors thank all the researchers and teams whose efforts have made possible the wide range of data sources used in this study, including InSAR, tide-gauge, altimetry, GNSS observations, and the GIA model.

## Author contributions

J.O., R.N., D.L., M.M., and L.S. designed research; J.O. performed research; J.O., R.N., D.L., M.M., M.S., L.O., B.H., and L.S. analyzed data; and J.O., R.N., D.L., M.M., M.S., L.S, L.O., D.D., J.H., B.H., and F.S. wrote the paper.

## Funding

## Competing interests

The authors declare no competing interests.
