## [Transparent Peer Review file · Nature Communications]

Subsidence more than doubles sea-level rise today along densely populated coasts

Corresponding Author: Dr Julius Oelmann

Version 0:

Reviewer comments:

Reviewer #1

(Remarks to the Author)

Review on

"Subsidence more than doubles climate-driven sea-level rise today along densely populated coasts"

By Julius Oelmann, Robert J. Nicholls, Daniel Lincke, Marta Marcos, Manoochehr Shirzaei, Laura Sánchez, Leonard Ohenhen, Denise Dettmering, Jochen Hinkel, Benjamin P. Horton, Florian Seitz

The manuscript is concerned with impact of subsidence on relative sea-level rise (RSL) and aims at providing "the most robust estimates of contemporary RSL change with the available data" . This topic is, no doubt, potentially interesting for a wide range of audiences. I find the manuscript overall well-written and potentially interesting for Nature Communications. However, some necessary details about uncertainties are missing. I recommend the manuscript to be considered after a major revision.

Specific comments

- Fig.1 should be accompanied by a figure of uncertainties of the VLM estimates. Is it Fig. S6C ? The statistical significance of trends is then to be discussed.

- Generally , I find the authors to be rather optimistic by displaying the average rates in the largest deltas, or elsewhere, to one or two decimal places (e.g. Table S2). For example, the authors report the Nile delta to subside at an average rate of -7.8 mm/year. What is about its uncertainty ? What is the VLM predicted by OE24 along the Nile delta coast ? Is it much different from the inSAR estimate there?

- VLM can drastically change on very different space and time scales. The authors combine the linear trend estimates of the three data sources (OE24 + InSAR + GNSS + GIA) in somewhat hierarchical manner. First, they map, where available, inSAR data, then, GNSS, VLM and, finally, GIA model. It would be interesting to see the coastal segments covered by inSAR dataset, those by GNSS, OE24 and by GIA.

- Did you compare inSAR measurements with OE24 and GIA model where all the data sets are available ?

- The authors mention "low confidence in VLM estimates" existing today. They are right. I believe the manuscript would benefit from a clearer distinguishing of limitations in the datasets they used. For example, inSAR data allow spatial averaging that is contrasted with point-wise nature of tide gauge records or single GNSS stations. On the other hand, tide gauge records are much longer than inSAR data. As to GNSS , they can be strongly influenced by compaction and, thus, by the depth of building on which the GNSS station is installed.

Discussing in more detail the space/time differences in VLM signals recorded in different datasets used by the authors would be beneficial for the paper as well as a careful uncertainty analysis explained in a manner understandable for large audience of Nature Communications.

Reviewer #2

(Remarks to the Author)

The authors aim to improve the integration of coastal subsidence with sea-level change and provide a global assessment that integrates the best-available datasets on VLM (or assumed VLM) through a hierarchical workflow to arrive to an integrated VLM dataset. This is then compared to (linear) sea level change data and used to compute the amount of (average) relative sea-level rise people living along the world's coasts are experiencing. This work is in itself highly relevant and may become a showcase on how to integrate various VLM data. While in general the authors take great care to discuss the results and limitations of the study (and also provide already a level of error propagation in the results), there are several aspects in the study which need to be considered to make the work more robust and to have the conclusions of the study rightfully underpinned. Several of the current workflow and data processing decisions may lead to an overestimation of VLM attribution to population, while at the same time potentially underestimating SLC. These issues are mendable and when properly dealt with this paper deserves publication at a level of Nature Comms, as I expect this paper to become influential in the community.

1) In their MS the authors directly infer VLM from InSAR. This is a common practice in which InSAR-estimated elevation change between satellite passes is directly interpreted as VLM, while it is an estimate of reflector distance difference (may or may not be caused by a motion). Assuming InSAR-estimated rates as VLM is an assumption at several levels (e.g. the observed difference in reflector distance is caused by VLM), but also that the observed InSAR values (which also go through a number of processing and post processing steps) indeed provide an estimate of VLM. This study is not processing InSAR but uses existing InSAR datasets so there is no first-handed processing, however for the clarity and robustness of the workflow, clearly considering the involved assumption – and their potential impacts on the results of this study – are of importance in this manuscript. Moreover as I expect this to be an exemplar study which will be followed by others in the future – this is the moment to set the bar.

2) Apart from the above mentioned assumptions in the methods to integrate the various datasets there are several decisions made (e.g. by applying a spatial averaging of InSAR results), which may influence the results of the study and which need further explanation/quantification/justification.

In the workflow presented by the authors, the InSAR results (as far as can be judged from the methods, data transformation for the different InSAR datasets is not explicitly written for each individual dataset. However in the combination section in the methods I found: "First, we use all InSAR datasets and interpolate them on the high-resolution grid of DIVA". The interpolation method is not specified (e.g. average?, median?).

In the InSAR Deltas dataset section the interpolation methods is described. Here the authors choose to use the spatial average, instead of the median. By following this practice, assumably also for the other InSAR datasets, the workflow itself has the tendency to overestimate the average experienced VLM per bulk population number. Especially for areas which experience high rates and high spatial variability in subsidence rates (e.g. deltas, cities with high differential subsidence) applying a spatial average is likely skewing the results towards higher rates for the entire segment (see e.g. Thieblémont et al., 2024). Why not use the median instead to avoid this skewing effect and potential overprediction of subsidence for an entire segment? To investigate the impact of this potential skewing effect, please provide a comparison using a median interpolation method as well (for the delta InSAR but also the other InSAR datasets). As such the authors will be able to quantify the potential spatial 'over-prediction' of higher subsidence for larger areas due to the skewing effect which may be caused by very local phenomena (e.g. a certain part of a city may sink very fast, e.g. northern harbour in Jakarta, which then causes a much larger part of the city's residents to experience higher VLM rates, which in reality they do not experience). Beside using the median to do a spatial check, it would be valuable to provide some examples for selected individual cities with high differential subsidence, using the original, high-resolution InSAR dataset and high resolution population data, in comparison to the results global applied approach – this will enable to substantiate/evaluate/provide a measure of uncertainty to the later results on high average VLM for coastal population at global level.

A similar question for interpolation quantification arises for the resampling of high-resolution grids to lower-resolution grids "Line 483: These segments are 482 then grouped/averaged into the lower-resolution grid (12,148 segments)." A comparison to median transformation would settle this.

Lines 152-158. I appreciate the discussion here on the limitations of statistical averages of VLM with respect to the real spatial variability within a delta. In the sentences that follow it becomes clear that the authors treat InSAR rates as a value of elevation change (VLM+sediment accretion). "It should also be noted that the reported InSAR rates usually represent total elevation change in non-urban dynamic landscapes, which is the sum of vertical land motion and vertical accretion, except for the Mississippi delta, where we account for vertical accretion using a dedicated estimate of Nienhus et al., 2020". This conclusion is in contrast to the earlier discussed assumption that InSAR reflect VLM – here it says it does not. While the authors are aware of the considerations to be made as they do discuss it (e.g. 310-323), the implications also need to be considered (or potential consequences at least made explicit) in the workflow decisions and result presentation.

In addition, the ability of InSAR to measure elevation change is an assumption on its own, which may very well not be the case. 1) the ability of InSAR to properly capture sediment accretion in an active sediment depositional setting is far from certain – especially when accretion rates are in the order of 1-2 mm/yr, which is lower than expected InSAR accuracy, especially in densely vegetated settings. The InSAR used (depending on type and processing of the InSAR) may/may not

include vertical accretion to some extent – although to which extent remains uncertain for such large scale studies with limited validation on their effectiveness to register accumulation under different circumstances, environments and vegetation types. 2) many of the studied deltas are densely populated, which means that any human-made structure (roads, buildings, grave stones etc) acts as reflectors for the InSAR signal, and per definition do not capture sedimentation effects. As the InSAR that is used here is processed into spatial pixels (in the process merging the response of different reflectors into a single pixel). Depending on the specific local situation, the InSAR rates provide a ‘smoothed’ signal or VLM reflectors (either with or without foundations and insensitive to sedimentation) and, potentially, elevation change observations.

Line 156: “except for the Mississippi delta, where we account for vertical accretion using a dedicated estimate of Nienhus et al., 2020. “ This is confusing and the decision is not explained. Only when reading the methods it becomes somewhat more clear why the authors did this. This is the only delta in which InSAR was substituted with a RSET derived map (thereby excluding the vertical accretion). I suggest adding more explanation here, as this will also help to provide a better underpinning of the following sentence:

Line 157 “ Therefore, in most cases these estimates represent lower bounds of actual subsidence.” It is unclear what the InSAR data provides for such environment (see above) and requires a more thorough consideration and substantiation – as it may not be the case – e.g. InSAR in wetland areas mainly reflects human structures which may be sinking faster than the surrounding wetlands due to additional loading effects.

Line 170 “ although this cause is speculative “ I appreciate this addition, and agree with this.

3) Uncertainty propagation of individual dataset into the final results

The authors go to great lengths to quantify the errors for the InSAR datasets in relation to GNSS results (showing that the overall error for the different InSAR datasets floats somewhere between 1-1.5 mm/yr). In addition, the authors argue that the InSAR datasets capture a large part of the coastal population (sinking cities/deltas etc). Other datasets may have similar errors/uncertainties involved). It would be useful to add a spatially explicit worldmap in addition to the maps of fig. 2, in which this uncertainty is shown, based on the underlying dataset used.

When I look at the data provided for the deltas (delta_VLM.xlsx), the error of the InSAR with the GNSS seem to be dominantly one-directional (InSAR subsidence > GNSS subsidence). Segments on average sink 1.25 mm/yr (0.85mm/yr median) faster than the GNSS rates. While this could be due to an under registration of the true rates by GNSS, it needs to be carefully considered/discussed, also in relation to potential overestimation stemming from spatial averaging practices.

The quantified errors/uncertainties are not reflected back in the final results and conclusions. With the above suggested error map – the authors will be able to substantiate their findings and instead of reporting a single value for VLM and amount of people experiencing this rate the uncertainty range can be provided (e.g. 307 million (+/- xxxx). For example, for Figure 2C an error envelope around the results. By providing a transparent and consistent propagation of errors/offsets in the processing framework, the results will be much more robust and future proof. In addition it will also help to pinpoint the sources of major uncertainties and set the agenda for future improvements.

4) The sea level change used in this study was inferred with a linear trend from recent observation starting from 1993. They authors argue this is only a few years before the VLM product (which one?). For sure this does not hold true for the INSAR, which is (I think all?) from the Sentinel 1 era (starting end of 2014). So there is a larger temporal discrepancy between the Sea level data and the area where InSAR is used. In addition, the authors use a linear function for sea level change, which in itself likely creates an underestimation of contemporary SLC (which is often non-linear). This approach likely underestimates contemporary sea level rise, but it is directly compared to (a.o.) InSAR-estimated VLM rates two decades younger, which likely (and unfairly) increasing the discrepancy between VLM and SLR. This approach needs to be re-evaluated, as most of the high VLM rates attributed to populations only covers the last decade – providing a better sea level change rate estimate that matches this time period will results in more fair comparison.

Minor issues:

Line 80: please clarify the 2D displacement mentioned here, this is confusing and it depends on how the InSAR is processed and whether or not combined with ascending/descending tracks – in principle providing a 3D observation of the movement (vertical, east-west, north-south). Perhaps here the authors have a specific InSAR processed product in mind.

Fig. 2c. discrepancy between caption and text in figure. 1-2-3 mm/yr in caption, in figure 0-1-2 mm/yr.

Figure S4/5. The statistical information related to the error bar (black lines) is missing.

Line 146 (and elsewhere): typo in Nienhuis reference.

Fig. 6D: Pop. Below 10m, add the 1e8 in this line to make it consistent with rest of the fig. – it is now hidden.

PM

Reviewer #3

(Remarks to the Author)

The paper presents a synthesis of globally available constraints on vertical land motion and sea surface elevation to provide a population-weighted measure of impact from relative sea level rise, focusing especially on major cities. The contributing datasets include InSAR, GNSS, tide gauges, altimetry and GIA models and sea surface satellite altimetry. The focus is on how the imaged relative sea level change impacts the human population. It will probably get some attention since sea level rise is of such high concern among the slow environmental risks facing society today. The paper is generally well written and seems to fit the appearance of articles published in Nature Communications in terms of length, voice, graphic quality, etc.

Overall it will be a nice contribution to the journal. The title is catchy, and it is interesting that the coastal human population tends to be concentrated in areas where VLM sign is on balance subsidence and are low lying. While this has been known for quite a while, the analysis quantifies the impact, sorts cities based on their exposure to subsidence, and draws out some interesting pieces of information about how sensitive the conclusions are to the constraints brought by different datasets (e.g., InSAR, GNSS, GIA models).

The dependence that InSAR has on GNSS is probably understated in the text. All InSAR analyses that pertain to vertical land motion of significance to sea level rise need to be aligned to geocentric reference frame in order to be compared to sea level trends. This is available only in the GNSS, and the InSAR relies on it completely. It was not immediately clear to me whether the difference InSAR data sources were aligned separately into the same GNSS reference frame. They appropriately showed level of agreement between GNSS and InSAR for each dataset in Table S1. The mentioned transforming the InSAR from ECMS with a latitude-dependent function, but did they do that for the other datasets too?

I make some more specific comments and suggestions below.

Detailed Comments

1. Not sure the phrase "climate-driven" needs to be in the title. Aren't all changes in sea level climate-driven since the level of the sea is a part of the whole climate system?

59. "Note that there are little or no human-induced uplift processes, so the effect of human-induced VLM acts overwhelmingly in one direction – subsidence". While this statement is technically true, I can think of one. The California Central Valley subsidence is implicated in driving hydrological mass loss that helps create unloading and uplift of the Sierra Nevada (Amos et al., 2014, Nature). But this effect is very slow, and the word "overwhelmingly" may appropriately capture the balance between anthropogenic and natural uplift.

83-94. Very long sentence. Could it be divided?

89. "in South, and Southeast, and East" would be better to say "in South, Southeast, and East" (link on line 307)

126-127. "fact that InSAR only covers about 17% of the global coastline by length, but almost 55% of the coastal population underlines its utility to resolve VLM for human and socio-economic analyses." It should be made clear to what extent this is because of data not being present (i.e., not covered by the satellite), versus data not being usable (e.g., decorrelation or other problem) versus data not processed/analyzed in the right way yet. Is it a matter of just more work needs to be done or is there a fundamental limitation to acquiring these data?

line 158. What is 'actual subsidence'? This phrase could be replaced with something more specific, such as 'vertical motion of solid earth'

272-273. This is a very long section header and should be shortened.

278-280. Getting repetitive.

314-315. Or vertical well extensometers.

321-323. The dependence that InSAR has on GNSS is probably understated here. All InSAR analyses that pertain to vertical land motion of significance to sea level rise need to be aligned to geocentric reference frame in order to be compared to sea level trends. This does not come across very well in the paper as it is now presented but should be stated explicitly here or in the Methods section.

363. "In conclusion, we advocate for a re-assessment of sea-level projections and impact assessment studies using a hybrid VLM data approach as presented here, incorporating all available data from different sources." Good statement but is not really the advertised conclusion of the paper... seems like the main point of the analysis was to emphasize the impact of VLM on human populations. The advocacy should come after the scientific conclusion, maybe put that sentence at the end of this paragraph.

376-379. This will make GNSS from NGL the underlying connection to the reference frame. Please state which reference frame the GNSS is in (e.g., ITRF14). NGL recently reprocessed all data in IGS2020 so either are available, and it is necessary to know which frame is used here.

422. "Blewitt et al., 2019" not in reference list. Probably what was meant was "Blewitt, G., W.C. Hammond, C. Kreemer, 2018, Harnessing the GPS Data Explosion for Interdisciplinary Science, Eos, 99, <https://doi.org/10.1029/2018EO104623>" ?

493. I clicked on the link to CMEMS and it made it to <https://data.marine.copernicus> but said the link is broken. Is there a more permanent link to cite?

559. Capitalization in references needs attention, many things that should be capitalized are not.

Figures

Figure 1. Move legends to side of map (not on top of) so as not to interfere with seeing some places in east and west Pacific

Not enough contrast between areas with 'black outlined markers' and other areas, e.g., most of major land masses seem to have a black outline around the colored symbol. Could that be made gray or some other color?

Figure 2. Figure 2C indicates what fraction of the population experience a given amount of VLM or less. It would seem more informative to flip the axis so the number represented was the fraction of the population that experiences that amount of subsidence or more.

Figure 2D colors an entire country based on what its coast is doing. While there may be policy consistency within a given country it does lead to false impressions of national importance of sea level rise in some cases. For examples, the Congo in Africa is virtually land locked, it only has ~50 km of coastline, so its exposure to risk may not be proportional to the area colored red. The US had a constant RSLC on the map whereas in reality its risk profile to RSLC is highly spatially variable.

Figure S2. What is the third row? very narrow, hard to see what this is. Does it add any insight or interpretive value to the data?

What is the value of "e" in the equation in the caption? The Thielblemont et al., 2024 citation is listed as in review.

This equation with latitude dependence is a very long wavelength correction for the difference between InSAR and GPS. Consequently, errors in the InSAR will still be present in shorter length scales (e.g. on the scale of a single swath of the satellite pass). Its possible that alignments are done within the InSAR products that are used as input to this analysis. If so please describe those details in the Methods section.

Figure S4. It looks like the red and blue colors grade towards white but there is no color bar to say what the different shades mean.

Figure S6B. What does the dot color mean? Horizontal axis needs annotation/labeling (caption says its population).

Version 1:

Reviewer comments:

Reviewer #1

(Remarks to the Author)

I think the comments were carefully and adequately considered by the authors and I recommend the manuscript for publication.

Reviewer #3

(Remarks to the Author)

The authors addressed my comments, I have only one remaining suggestion:

In Figure S8 there are three panels. First is absolute sea level change rate for the full period (a), second is ASLC rate since 2010 (b). The third is ASLC rate full period minus ASLC rate since 2010 (c). This results in areas that have accelerated ASLC rate appear negative in the third panel. Wouldn't you want it the other way around so it appears that positive changes in ASLC rate appear positively in the third panel? That seems to make more sense to me since a positive value suggests a positive acceleration in sea level rise.

Reviewer #4

(Remarks to the Author)

The authors produced an interesting synthesis of available vertical land motion data along populated coasts to be used for relative sea-level rise assessments. I think this is a very useful endeavor. I think what they produced is a critical step forwards, although like the authors mention, a VLM product should consist of more than only observations, it should also consist of subsurface information and drivers of subsidence. But, with respect to the large spatial scale of this study, I understand that cannot be achieved at this stage.

Throughout the manuscript I made some comments which can be easily addressed. Some major points:

- Please already mention in the Introduction that a VLM product based on observations is only half the story, the other half, which you don't use in the analysis, regards the use of subsurface data, subsurface use, policy etc. This aspect is now only briefly mentioned in the Discussion, but that is too late. We already know for years that VLM cannot be understood using geodesy alone, so please follow the state-of-the-art; there are numerous relevant studies to be found.

- Second, I read NI21b a while ago, but I can assure you, I don't know its details anymore. So please, give a brief summary in the beginning. That said, I am not entirely convinced that you should compare this study to NI21b at all. It looks like you are comparing apples with oranges (observations vs. expert judgement – why would you do that?). I also see that some of the NI21b authors are involved in this new research, so why the obsession of comparing yourself?
- I think making a global coastal VLM map is a proper scientific product. It's a product most people can understand. I cannot follow the subsequent steps in which you make a derivative of population-weighted VLM. I have no clue what it physically means or why it is relevant at all. Please reconsider that section or maybe shorten it and place it in the Discussion.
- Large parts of the Results section aren't scientific results, please check that and transfer sections to Discussion/Introduction.

Good luck, Kay Koster

Version 2:

Reviewer comments:

Reviewer #4

(Remarks to the Author)

This is fine now - no further comments

REVIEWER COMMENTS

Reviewer comments are shown in **bold**, author responses in regular font, manuscript text in Times New Roman, and changes to the manuscript are highlighted in red.

Reviewer #1 (Remarks to the Author):

Review on

"Subsidence more than doubles climate-driven sea-level rise today along densely populated coasts"

By Julius Oelmann, Robert J. Nicholls, Daniel Lincke, Marta Marcos, Manoochehr Shirzaei, Laura Sánchez, Leonard Ohenhen, Denise Dettmering, Jochen Hinkel, Benjamin P. Horton, Florian Seitz

The manuscript is concerned with impact of subsidence on relative sea-level rise (RSL) and aims at providing "the most robust estimates of contemporary RSL change with the available data" . This topic is, no doubt, potentially interesting for a wide range of audiences. I find the manuscript overall well-written and potentially interesting for Nature Communications. However, some necessary details about uncertainties are missing. I recommend the manuscript to be considered after a major revision.

Dear Reviewer,

We are very thankful for your comments. We have added a much more thorough analysis of uncertainties, including formal, cross-validation and spatial uncertainties. We have also improved the discussion of possible implications for the global average statistics, as well as possible influences of theoretical systematic biases in some of the InSAR datasets and their potential implications. We believe that this has greatly improved the paper, since it demonstrates which regions/datasets/uncertainties are most important for our global results.

Specific comments

- Fig.1 should be accompanied by a figure of uncertainties of the VLM estimates. Is it Fig. S6C ?

The statistical significance of trends is then to be discussed.

We adjusted Figure 1 and added subpanel d) showing uncertainties of the datasets.

Figure 1. Hybrid estimate of vertical land motion (VLM) along the global coastlines. The main map (a) shows uplift (positive signals) and subsidence in mm/year. Values outlined by black circles present regions where InSAR data is available, or where single GNSS estimates are used. The points are scaled by their absolute value to highlight strong subsidence or uplift. The pie charts on the left (b) and (c) show present the fraction of coastal population (length) covered by the individual datasets (InSAR, GIA, GPS, and the interpolated data form OE24). The colors depict the population- (or length-) weighted vertical land motion (in mm/year) of the individual datasets. (d) displays the estimated 1σ VLM uncertainties of the datasets (using the same color scale as in (a)), which are computed from the formal, spatial and cross-validation uncertainties in the InSAR data, and the provided uncertainties from OE24 and Caron et al., 2024 (see Methods).

We have also updated the text at multiple positions in order to improve the representation of uncertainties and propagate them into our main results (note that the following response may also answer several of the other comments). First, we revised the treatment of uncertainties as follows:

Methods L587: As discussed in the main text, uncertainties are caused by various causes: These can be broadly categorized into random errors and cross-validation uncertainties in InSAR and GNSS observations (e.g., temporal variability in local conditions, or random errors in geophysical corrections), which are ideally reflected in the provided formal trend uncertainties and by the

differences between the datasets; *systematic uncertainty* (biases, reference frame offsets between InSAR and GNSS), *methodological* (differences between processing approaches and physical models, e.g., GIA models or the VLM interpolation), *temporal and spatial uncertainty* (different window lengths, spatial variability and aggregation approach), as well as *model-input or parameter uncertainty* (e.g., GIA model Earth parameters). Thus, to understand the influence of these factors on our global averaged statistics (specifically the population-weighted global VLM), we compute the following uncertainty metric for all InSAR datasets:

$$\sigma = \sqrt{\sigma_{\text{formal}}^2 + \sigma_{\text{cross-validation}}^2 + \sigma_{\text{spatial}}^2}$$

...

Main text (L193): Non-linearities not only complicate extrapolations of the VLM signals, but they can also cause uncertainties in the global estimates due to inconsistent observation periods between observations (InSAR, GNSS, tide gauges). VLM uncertainties are also influenced by technique-dependent noise, differences in observation-window length, cross-validation uncertainties (e.g., potential offsets between InSAR and GNSS rates), spatial variability and aggregation effects, and parameter uncertainty. We integrate these different kinds of uncertainty in Figure 1d (see Methods, and SI Figs. S6d and S7). Median VLM uncertainties are generally higher in the most populated cities (2.6 mm/year) and deltas (1.8 mm/year) compared to all other regions (0.9 mm/year), which is largely controlled by the InSAR uncertainties (SI Fig. S6d). ...

Main text (second section, L272): Uncertainties in the global population-weighted mean subsidence (Fig. 2b) are influenced by the estimated VLM data uncertainties (contributing to ± 0.45 mm/year, Fig. 2c) and by the spatial heterogeneity of VLM and population (contributing to ± 0.32 mm/year). The choice of the averaging method also affects the global statistics: Using median InSAR rates instead of means when aggregating the data from the high- to the low-resolution DIVA grid (see Methods) reduces the population-weighted mean VLM of 0.27 mm/year (see also SI Fig. S7 for an overview). Since the population-weighted average VLM uncertainties of 3 mm/year (1σ) are substantially larger than the ASL trend uncertainties (0.9 mm/year, Fig. 2c), they dominate uncertainties in RSL estimates. However, the fundamentally different approaches to define uncertainty in the VLM still hinder an objective comparison and therefore require consistent frameworks to assess these differences.

We also added SI Figure S7 to illustrate the effect of random normally distributed errors, and hypothetical biases in different datasets on the global mean statistics.

- Generally , I find the authors to be rather optimistic by displaying the average rates in the largest deltas, or elsewhere, to one or two decimal places (e.g. Table S2). For example, the authors report the Nile delta to subside at an average rate of -7.8

mm/year. What is about its uncertainty ? What is the VLM predicted by OE24 along the Nile delta coast ? Is it much different from the InSAR estimate there?

We fully agree, as delta subsidence is highly variable (also discussed by Ohenhen et al., 2025), and simple averages may sometimes not be very representative of this variability. Other limitations are that some deltas, such as the Nile delta, are very poorly covered by GPS, which hinders any meaningful comparison in terms of delta average subsidence. All these differences are provided in the additional overview of differences between VLM estimates from InSAR data with GPS trends, the results from Nicholls et al., (2021), and multi-technique weighted averages in Figure S5 and the appended SI table (i.e., the data file). There are also differences in terms of the aggregation of data: For instance, in the text and in the in appended SI table we report delta averages over the **entire delta** (as provided by Ohenhen et al., 2025), these averages can however differ from the averages of the **coastal values** mapped onto the DIVA grid. For example, the delta average for the *entire* Nile delta is -7.8 mm/year (InSAR), the *coastal* average is 7 mm/year, the single GPS station that is within 50km to the coast shows a trend of -2.5 mm/year. The closest InSAR estimate to that stations is -2.6 mm/year. The average of all coastal estimates of the interpolated VLM data (OE24) are 2.2 mm/year, reflecting it's dependence on the limited GPS data in that region (please also refer to SI Figure S2 for a more comprehensive overview of the comparison InSAR vs. GPS). Note that the averaged InSAR data in the table are thus not directly comparable with the GPS data:

Methods (L530): More information on the estimates in deltas can be found in the SI data file 1 and Figure S5, where we show information for all deltas (following the deltas listed in Ericson et al., 2006, Table 2), where any geodetic information is available. Here, we report delta averages from InSAR (averaged over the entire delta surface, or averaged over the InSAR data points, that are interpolated onto the DIVA segments), as well as from GNSS, and a combination of InSAR and GNSS. Note, that these estimates sometimes differ substantially, which is mainly due to the large spatial variability of VLM, and the heterogeneous coverage of GNSS stations in the deltas.

Accordingly, these results cannot be directly compared as they do not represent co-located (InSAR-GNSS) estimates. In SI data file 1, we also report GNSS estimates for deltas where no InSAR data is available.

We also changed the main text (L163): We find that most of the largest deltas are subsiding with average rates (and spatial standard-deviations) of -5.4 (3.6) mm/year in the Ganges/Brahmaputra delta, -7.8 (2.9) mm/year in the Nile delta, -2.7 (2.7) mm/year in the Yangtze delta, and -5.6 (5.5) mm/year in the Mekong delta (see SI Fig. S5). While these statistical averages (over the entire deltas) are useful for a general overview of delta subsidence, they do not represent the substantial spatial variability of subsidence within the deltas, which can reach values of up to 12 mm/year (95th percentile) in the Ganges or Nile delta, for instance. Hence, the new InSAR estimates are crucial for assessing these variations, which was so far hindered by the poor coverage by GNSS stations (SI Fig. S5).

Overall, given the complex delta subsidence processes, delta subsidence still represents one of the major sources of uncertainties (also shown in SI Fig. S7), however, these new InSAR estimates are a crucial step towards improving our understanding of these variations. Hence we also added in the discussion (L355):

However, delta subsidence estimates, or InSAR VLM estimates in general, are still subject to several sources of uncertainty. Next to the discussed formal, spatial and cross-validation uncertainties, which are here treated as random non-biased effects, First, as mentioned above, the InSAR data used here typically represent total surface elevation change and do not distinguish between shallow and deep subsidence or vertical accretion. it remains uncertain whether InSAR measurements fully capture the combined effects of shallow and deep subsidence, or to what extent they are influenced by vertical accretion processes. These factors can be differentiated using in situ data (rod surface elevation table (RSET) and marker horizon records, e.g., Nienhuis et al., 2017, Santillan et al., 2023, or vertical well extensometers), which is done for the Mississippi Delta. However, such local information is not available on a global scale, which may also contribute to differences with respect to NI21b, who also aggregated data based on such in situ information. ~~Second, if~~ reflectors such as buildings have different foundation depths, this can lead to high spatial variability due to differential settlement between these reflectors. Buildings with deeper foundations are also less affected by shallow subsidence. This could cause an overall underestimation of the population-weighted average subsidence, since it strongly depends on InSAR estimates from the most densely populated areas and may also contribute to differences w.r.t. NI21b [Minderhoud et al., 2025]. Expectedly, we find that the global population-weighted average subsidence (as provided in Table 1) is most sensitive to possible biases in the city and delta datasets: A hypothetical bias of 1 mm/year in either of these datasets would translate to a ~0.3 mm/year bias in the global averaged VLM.

Finally, we now also only report VLM rates for deltas to the first decimal place.

- VLM can drastically change on very different space and time scales. The authors combine the linear trend estimates of the three data sources (OE24 + InSAR + GNSS + GIA) in somewhat hierarchical manner. First, they map, where available, inSAR data, then, GNSS, VLM and, finally, GIA model.

It would be interesting to see the coastal segments covered by inSAR dataset, those by GNSS, OE24 and by GIA.

We appreciate this suggestion and have added information about how the data is combined in a map in Figure S9. We also strongly improved the description of how the data is combined/how uncertainties are computed. Note, that we now also include InSAR data for New Zealand.

a

Figure S9: Global distribution of the underlying datasets used in the VLM estimate.

- Did you compare InSAR measurements with OE24 and GIA model where all the data sets are available?

We did not compare the InSAR measurements directly with OE24 and GIA. The main reason is simply because the GIA model doesn't resolve any other process and OE24 does not resolve the high-resolution local information provided by InSAR (in particular in SE-Asia). Therefore, we have only compared InSAR with point-wise GNSS data for the US, Europe, and cities, see SI Fig. S2. This comparison shows an agreement with InSAR and GNSS data, which is usually in the order of 1-1.5 mm/year (for the USA and Europe).

Comparing these datasets with GIA would be most meaningful at locations where we have a relatively strong GIA signal. Such a comparison was done for the US (Ohenhen et al., 2024) from the SI file https://static-content.springer.com/esm/art%3A10.1038%2Fs41586-024-07038-3/MediaObjects/41586_2024_7038_MOESM1_ESM.pdf:

Fig. 3. Distribution of glacial isostatic adjustment (GIA) effects on vertical land motion (VLM) for selected US coastal cities. (a) Boxplots representing the distribution of total VLM and VLM without GIA for 11 US Atlantic coastal cities. **(b)** Boxplots representing the distribution of total VLM and VLM without GIA for 11 US Gulf coastal cities. **(c)** Boxplots representing the distribution of total VLM and VLM without GIA for 11 US Pacific coastal cities.

They demonstrate that InSAR often captures local subsidence not fully explained by GIA alone (from Ohenhen et al., 2024): ‘As GIA is the main natural driver, we used the GIA ICE-6G-D model⁷⁰ to estimate the GIA contributions at the SAR pixels and subtracted its effect from the observed VLM to assess the non-GIA contributions to the estimated VLM along US coasts (Supplementary Fig. 2). The relative reduction of subsidence by 46%, 4% and 20% for the Atlantic, Gulf and Pacific coasts, respectively, suggests that the effect of GIA on subsidence is dominant primarily along the US Atlantic coast and minimal for the Gulf and Pacific coasts (Supplementary Fig. 2c–e). Although the median rates of subsidence are reduced for all 32 major coastal cities, several areas with subsidence rates greater than 2 mm per year remain apparent in more than half of the selected cities, such as Boston, Atlantic City, Charleston, Biloxi, New Orleans, Texas City, San Francisco, Foster City and San Diego (Supplementary Fig. 3).’

We find a quite similar result for Europe: While the large scale, GIA fingerprint agrees across the datasets, small scale subsidence is usually not resolved by the smoother interpolated dataset OE24 nor the GIA model:

We hope that the validation provided in the appendix and the statistics on all InSAR datasets (comparison of trends v.s. GNSS) is sufficient to make judgements about its accuracy.

- The authors mention "low confidence in VLM estimates" existing today. They are right. I believe the manuscript would benefit from a clearer distinguishing of limitations in the datasets they used. For example, inSAR data allow spatial averaging that is contrasted with point-wise nature of tide gauge records or single GNSS stations. On the other hand, tide gauge records are much longer than inSAR data.

As to GNSS , they can be strongly influenced by compaction and, thus, by the depth of building on which the GNSS station is installed.

We have added more explanations on the remaining limitations and differences across datasets. As explained in more detail in the previous comment, we have improved the discussion of error sources stemming from the spatial averaging, the different observation lengths, technique-dependent noise, and the dependence of foundation depth on the extent to which shallow subsidence or compaction is observed (e.g., in L193, L264, L272, L353, ...). As mentioned by the reviewer, inconsistencies remain, because of the different methods used to compute uncertainties for the different techniques, including differences in window length. Therefore, we added (L619): It is important to emphasize that this remains a

simplified way of combining uncertainties derived from fundamentally different methods. A more rigorous treatment would require holistic probabilistic frameworks capable of jointly modeling technique-dependent noise and biases, while also resolving the temporal evolution of height changes, which is challenging especially for the very large InSAR datasets. Such an approach could help to mitigate inconsistencies arising from differing observation windows.

Please also refer to our changes in the discussion:

This also underlines the importance of global networks of dense and continuous GNSS measurements (e.g., as provided by the Nevada Geodetic Laboratory, Blewitt et al., 2018), which are the prerequisite to constrain InSAR VLM estimates to a global reference frame, and are thus fundamental to reduce biases in regional estimates. In addition, when GNSS stations are used as reference stations and have a deep anchor depth, they will miss shallow compaction and thus lead to an underestimation of total subsidence.

Discussing in more detail the space/time differences in VLM signals recorded in different datasets used by the authors would be beneficial for the paper as well as a careful uncertainty analysis explained in a manner understandable for large audience of Nature Communications.

We really appreciate this comment and have therefore strongly improved the treatment of uncertainties (accounting for spatial, temporal and cross-validation uncertainties). We hope that most of the information provided before addresses this comment already. To avoid repetitions, we highlight here only the key revisions related to clarifying the treatment of uncertainties, particularly those arising from differences in observation periods (I184):

Subsidence can additionally be highly non-linear [e.g., Cao et al., 2021, Shirzaei et al., 2021, Törnqvist and Blum, 2024], for example, due to both rapid acceleration in subsidence caused by excessive groundwater extraction and rapid cessation of subsidence when excessive groundwater extraction is stopped, as in Tokyo and Osaka in the 1960s/70s (Fig. 2 in Cao et al., 2021). Therefore, we note that, since InSAR based VLM rates are generally derived from a relatively short period of measurements (~5-15 years), they only provide a snapshot of the present linear rates, which may change in future. Such non-linear changes, and the drivers of subsidence need to be better understood and quantified and such investigations can be supported by this combined dataset (see Discussion). Non-linearities not only complicate extrapolations of the VLM signals, but they can also cause uncertainties in the global estimates due to inconsistent observation periods between observations (InSAR, GNSS, tide gauges). VLM uncertainties are also influenced by technique-dependent noise, differences in observation-window length, cross-validation uncertainties (e.g., potential offsets between InSAR and GNSS rates), spatial variability and aggregation effects, and parameter uncertainty. We integrate these different kinds of uncertainty in Figure 1d (see Methods, and SI Figs. S6d and S7).

Finally, because uncertainties are indeed the largest in the most critical regions we added in the conclusions (L431): Focusing future improvements on the most heavily populated and rapidly subsiding coastal regions is essential, also because VLM estimates in these areas remain the most consequential and yet are among the most uncertain.

Reviewer #2 (Remarks to the Author):

The authors aim to improve the integration of coastal subsidence with sea-level change and provide a global assessment that integrates the best-available datasets on VLM (or assumed VLM) through a hierarchical workflow to arrive to an integrated VLM dataset. This is then compared to (linear) sea level change data and used to compute the amount of (average) relative sea-level rise people living along the worlds coasts are experiencing. This work is in itself highly relevant and may become a showcase on how to integrate various VLM data. While in general the authors take great care to discuss the results and limitations of the study (and also provide already a level of error propagation in the results), there are several aspects in the study which need to be considered to make the work more robust and to have the conclusions of the study rightfully underpinned. Several of the current workflow and data processing decisions may lead to an overestimation of VLM attribution to population, while at the same time potentially underestimating SLC. These issues are mendable and when properly dealt with this paper deserves publication at a level of Nature Comms, as I expect this paper to become influential in the community.

Dear PM,

We are very thankful for your comments, which have led to several improvements in the manuscript. The main change is an improved uncertainty analysis: We investigate now the influence of formal, spatial and cross-validation uncertainties in the InSAR datasets. We also more accurately discuss possible biases in the InSAR data, for example, due to different foundation depths of reflectors. We show the relative importance of uncertainties in the individual datasets and how hypothetical biases could alter global statistics. In addition, we discuss the effects of averaging (mean vs. median) and compare all these different uncertainty sources associated with the VLM estimation.

We have also clarified that the climate-driven absolute sea level change is currently the main driver of global population-weighted RSL, and that it may be even more important in future, assuming no further acceleration in the global pop.-weighted subsidence rates. Please note that as natural climate variations become more dominant on shorter timescales in the altimetry-based trends, we believe keeping the current time window, while still highlighting the ongoing and future acceleration (please refer to more details in the comments) is a reasonable choice. We hope that the following responses address the comments satisfactorily and make the paper more robust.

1) In their MS the authors directly infer VLM from InSAR. This is a common practice in which InSAR-estimated elevation change between satellite passes is directly interpreted as VLM, while it is an estimate of reflector distance difference (may or may not be caused by a motion). Assuming InSAR-estimated rates as VLM is an assumption at several levels (e.g. the observed difference in reflector distance is caused by VLM), but also that the observed InSAR values (which also go through a number of processing and post processing steps) indeed provide an estimate of VLM. This study is not processing InSAR but uses existing InSAR datasets so there is no first-handed processing, however for the clarity and robustness of the workflow, clearly considering the involved assumption – and their potential impacts on the results of the this study – are of importance in this manuscript. Moreover as I expect this to be an exemplar study which will be followed by others in the future – this is the moment to set the bar.

We have changed the text at several positions to better explain what InSAR actually measures, and what the assumptions and remaining limitations are:

L81: While these techniques represent sparse point-wise measurements, InSAR provides 2D-can provide velocity estimates in satellite's line of sight for a large number of coherent reflectors, such as buildings or infrastructure. Using known ground velocities, typically derived from GNSS measurements referenced to a global reference frame, these line-of-sight velocities can be transformed into displacement two-dimensional VLM estimates at a resolution of several meters ~~and is~~. Accordingly, InSAR is being increasingly used in many large coastal cities or more widely

First section (L116): To refine the resolution of VLM in major coastal cities and the largest deltas, where VLM is often largest, we exploit InSAR VLM data from the European Ground Motion Service (EGMS, see also Thiéblemont et al., 2024), and from Hamling et al., 2022, Ohenhen et al., 2024, 2025, Shirzaei et al., 2024, and Ao et al., 2024, which resolves subsidence at very small scales, and in the most populated areas where no GNSS (or tide gauge) station data are publicly available. Here, our assumption is that the InSAR rates are reflecting VLM. This is supported by the estimated accuracies of the datasets from validations with GNSS measurements, which are usually between 1-2 mm/year (see also Methods, Fig. S2 and Table S1). Nonetheless, several factors and potential biases may violate this assumption, and are therefore discussed in detail in the main article.

L201: ... It should also be noted, that there is uncertainty regarding the extent to which InSAR-derived subsidence rates may be underestimated, either because shallow subsidence is only partially observed (when reflectors such as buildings are founded on deeper layers), or due to the influence of vertical accretion in dynamic, non-urban landscapes [Törnqvist and Blum, 2024, Minderhoud et al., 2025]. ...

We also added a much more thorough uncertainty quantification and a discussion of possible biases in the datasets (as explained in more detail in the comments below).

2) Apart from the above mentioned assumptions in the methods to integrate the various datasets there are several decisions made (e.g. by applying a spatial averaging of INSAR results), which may influence the results of the study and which need further explanation/quantification/justification.

In the workflow presented by the authors, the InSAR results (as far as can be judged from the methods, data transformation for the different InSAR datasets is not explicitly written for each individual dataset. However in the combination section in the methods I found: “First, we use all InSAR datasets and interpolate them on the high-resolution grid of DIVA”. The interpolation method is not specified (e.g. average?, median?).

We have improved the method section ‘Combination’. We have now also computed the medians in addition to the mean when averaging the data from the HR (250k-segment DIVA grid) to the final (12k-segment) grid (L572).

(1) First, we use all the VLM trends and (1σ) formal uncertainties of all InSAR datasets (China [Ao et al., 2024], coastal cities, New Zealand, USA, Europe, Delta) and interpolate them with nearest neighbour interpolation on the high-resolution grid of DIVA (consisting of 247,666 segments), wherever available. These segments are then grouped/averaged/averaged -intoto the lower-resolution grid (12,148 segments) by computing both mean and median values within each cell.- In that process, we also compute the standard deviation of all averaged VLM trends, to estimate spatial uncertainty $\sigma_{spatial}$.

The median results are now shown in Figure S3 (dashed line) and Figure S7 (shown in another comment below):

As can be seen, the global averages based on median InSAR data are slightly lower (as suggested by the reviewer in the next comment) indicating the influence of single extreme values on the mean InSAR VLM estimates. Accordingly, we added in the text L194:

Uncertainties in the global population-weighted mean subsidence (Fig. 2b) are influenced by the estimated VLM data uncertainties (contributing to ± 0.45 mm/year, Fig. 2c) and by the spatial heterogeneity of VLM and population (contributing to ± 0.32 mm/year). The choice of the averaging method also affects the global statistics: Using median InSAR rates instead of means when aggregating the data from the high- to the low-resolution DIVA grid (see Methods) reduces the population-weighted mean VLM of 0.27 mm/year (see also SI Fig. S7 for an overview).

Note that we will also publish the entire code base (on github) that describes how these estimates are derived. Please refer to the comments below that better explain the new treatment of uncertainties.

In the InSAR Deltas dataset section the interpolation methods is described. Here the authors choose to use the spatial average, instead of the median. By following this practice, assumably also for the other InSAR datasets, the workflow itself has the

tendency to overestimate the average experienced VLM per bulk population number. Especially for areas which experience high rates and high spatial variability in subsidence rates (e.g. deltas, cities with high differential subsidence) applying a spatial average is likely skewing the results towards higher rates for the entire segment (see e.g. Thieblémont et al., 2024). Why not use the median instead to avoid this skewing effect and potential overprediction of subsidence for an entire segment? To investigate the impact of this potential skewing effect, please provide a comparison using a median interpolation method as well (for the delta InSAR but also the other InSAR datasets). As such the authors will be able to quantify the potential spatial ‘over-prediction’ of higher subsidence for larger areas due to the skewing effect which may be caused by very local phenomena (e.g. a certain part of a city may sink very fast, e.g. northern harbour in Jakarta, which then causes a much larger part of the city’s residents to experience higher VLM rates, which in reality they do not experience). Besides using the median to do a spatial check, it would be valuable to provide some examples for selected individual cities with high differential subsidence, using the original, high-resolution InSAR dataset and high-resolution population data, in comparison to the results of a global applied approach – this will enable to substantiate/evaluate/provide a measure of uncertainty to the later results on high average VLM for coastal population at global level.

A similar question for interpolation quantification arises for the resampling of high-resolution grids to lower-resolution grids “Line 483: These segments are 482 then grouped/averaged into the lower-resolution grid (12,148 segments).” A comparison to median transformation would settle this.

We fully agree with this comment and have in response also computed median InSAR estimates which indeed shows slightly reduced rates in the global mean statistics (please also refer to our previous response).

We also agree that uncertainties arising from the interpolation, and the InSAR and population data distribution are not yet incorporated. Thus, in order to better account for uncertainties in the InSAR data, and in the uncertainty about which population fraction is covered by different InSAR estimates, we have integrated an uncertainty metric that integrates formal, spatial and cross-validation uncertainties of the InSAR data (please note that some of these changes are repetitions of the responses to the first reviewer). Main text L193:

Non-linearities not only complicate extrapolations of the VLM signals, but they can also cause uncertainties in the global estimates due to inconsistent observation periods between observations (InSAR, GNSS, tide gauges). VLM uncertainties are also influenced by technique-dependent noise, differences in observation-window length, cross-validation uncertainties (e.g., potential offsets between InSAR and GNSS rates), spatial variability and aggregation effects, and parameter uncertainty. We integrate these different kinds of uncertainty in Figure 1d (see Methods, and SI Figs. S6d and S7). Median VLM uncertainties are generally higher in the most populated cities (2.7

mm/year) and deltas (1.8 mm/year) compared to all other regions (0.9 mm/year), which is largely controlled by the InSAR uncertainties (SI Fig. S6d).

Methods L587:

As discussed in the main text, uncertainties are caused by various causes: These can be broadly categorized into *random errors* and *cross-validation* uncertainties in InSAR and GNSS observations (e.g., temporal variability in local conditions, or random errors in geophysical corrections), which are ideally reflected in the provided formal trend uncertainties and by the differences between the datasets; *systematic uncertainty* (biases, reference frame offsets between InSAR and GNSS), methodological (differences between processing approaches and physical models, e.g., GIA models or the VLM interpolation), *temporal and spatial* uncertainty (different window lengths, spatial variability and aggregation approach), as well as *model-input or parameter* uncertainty (e.g., GIA model Earth parameters). Thus, to understand the influence of these factors on our global averaged statistics (specifically the population-weighted global VLM), we compute the following uncertainty metric for all InSAR datasets:

$$\sigma = \sqrt{\sigma_{time,random}^2 + \sigma_{cross-validation}^2 + \sigma_{spatial}^2}$$

Here, $\sigma_{time,random}^2$ are assumed to be represented by the provided formal uncertainties of the different products, as both window-length and noise amplitude influence formal uncertainties. Note that although the reported formal uncertainties may not be fully comparable across datasets, as they are derived using different methodologies, the combined uncertainties are usually dominated by the systematic and spatial components. Next, we use the standard deviations of the differences w.r.t. GNSS as *cross-validation* uncertainties $\sigma_{cross-validation}^2$ for each dataset, which are usually within the 1-2 mm/year ranges. Finally, because the data is aggregated from a high (~250k) to a low (~12k) resolution grid, we use the standard deviation of trends at locations that are aggregated to single points on the low resolution grid as a measure of *sampling/spatial* uncertainty $\sigma_{spatial}^2$. Thus, we assume that the data follow a normal distribution. While this is a simplification, it enables a more straightforward combination with the other uncertainty components and avoids the need to estimate multiple distribution parameters for each location. For consistency, we thus estimate the spatial standard deviations (uncertainties) for the Chinese cities based on the interquartile ranges (divided by ~1.35) from the derived non-normal Johnsons S_H distributions. Note that since we are not matching every InSAR pixel with high resolution population information, this spatial uncertainty should also express the uncertainty about the high-resolution distribution of people that are also aggregated on the lower resolution grid.

Uncertainties at all other locations where no InSAR is available are based on the probabilistic trend uncertainties of the interpolated data [OE24], that combines temporal, spatial and technique dependent noise, formal GNSS uncertainties (Blewitt et al., 2018), as well as the model uncertainties that are based on a large GIA model ensemble from Caron et al., 2018 (and thus accounts for earth-parameter uncertainty). It is important to emphasize that this remains a simplified way of combining uncertainties

derived from fundamentally different methods. A more rigorous treatment would require holistic probabilistic frameworks capable of jointly modeling technique-dependent noise and biases, while also resolving the temporal evolution of height changes, which is challenging especially for the very large InSAR datasets. Such an approach could help to mitigate inconsistencies arising from differing observation windows.

To compute the uncertainty in the population-weighted mean statistics, we also account for uncertainties associated with the spatial distribution or sampling of data using bootstrapping (Fig. 3 and SI Fig. S7). We compute the population-weighted mean for 10,000 randomly selected subsets of VLM estimates, accounting for 50% of the total sample size (i.e., the size of the low-resolution grid 12,148). This uncertainty expresses the sensitivity of the global mean to the heterogeneity of the distribution of VLM rates and population sizes and explores if the mean is also significant for subsets of the data. This uncertainty is larger when the global average statistic is controlled by only a few locations, and vice versa. For each resampled subset, we also perturb the VLM trends by randomly distributed errors drawn from normal distributions, which are scaled by the derived uncertainty metrics (as the standard deviations of the distributions) of the techniques, as explained before. The influence of the resampling, the random perturbations, the different InSAR averaging approaches (mean vs. median), and hypothetical biases (representing systematic uncertainties) in each InSAR dataset are shown in the SI Figures S6d and S7.

As described, by including the spatial variability, we account for the uncertainties in the high resolution distribution of the population and the potential errors/mismatches that can emerge when aggregating the data to the lower resolution DIVA grid. Accordingly, we discuss causes of uncertainty in the global mean pop.-weighted VLM in the text.

Uncertainties in the global population-weighted mean subsidence (Fig. 2b) are influenced by the estimated VLM data uncertainties (contributing to ± 0.45 mm/year, Fig. 2c) and by the spatial heterogeneity of VLM and population (contributing to ± 0.32 mm/year). The choice of the averaging method also affects the global statistics: Using median InSAR rates instead of means when aggregating the data from the high- to the low-resolution DIVA grid (see Methods) reduces the population-weighted mean VLM of 0.27 mm/year (see also SI Fig. S7 for an overview). Since the population-weighted average VLM uncertainties of 3 mm/year (1σ) are substantially larger than the ASL trend uncertainties (0.9 mm/year, Fig. 2c), they dominate uncertainties in RSL estimates. However, the fundamentally different approaches to define uncertainty in the VLM still hinder an objective comparison and therefore require consistent frameworks to assess these differences.

This text often refers to SI Figure S7, where we compare the different contributions to the uncertainty in the global pop.-weighted mean VLM:

Figure S7: Contributions to uncertainties in the global population-weighted global VLM estimates. a) VLM values are perturbed using normally distributed random errors (see Methods) over 10,000 iterations. Shown are the resulting distributions together with the mean (solid lines), median (dashed lines), and the 15.9–84.1 percentile range (red lines, corresponding to ± 1 standard deviation for a normal distribution). b) Uncertainty arising from spatial heterogeneity is estimated by repeatedly computing population-weighted mean VLM from 10,000 random subsamples containing 50% of the original data. c shows the combined effects of subsampling and random perturbations. d) Distributions obtained when using *median* InSAR rates (instead of means, as shown in a-c) during the aggregation from high-resolution InSAR to the coarser DIVA grid, based on the same subsampling and perturbation approach as in panels a and b. e) Relative contributions of the different uncertainty sources to the uncertainty in the mean. We additionally illustrate how *hypothetical systematic biases of ± 1 mm yr⁻¹ in the dataset would affect the global mean VLM estimate, highlighting the sensitivity of the results to potential measurement biases.

Figure S6d shows now these combined data uncertainties:

Figure S6: a) shows the station density in terms of number of stations per 100² km. The station density was computed based on the number of GNSS stations (from NGL) within a 500 km search radius around an individual station, divided by the area covered by this circle. b) compares the station density versus population (on a logarithmic scale). Colorbar shows the kernel density estimate (KDE), indicating how densely points cluster in the (population, station density) space. The values of the station density and the population (defined at the coastlines) are coupled by nearest neighbor interpolation. c) shows the VLM uncertainties, the InSAR and GPS data are highlighted by blue outlines. The boxplots in d) present the VLM uncertainties of the different dataset components. Note that these uncertainties are based on the combined formal, spatial and cross-validation uncertainties in the InSAR datasets (and the provided uncertainties in OE24, GNSS (Blewitt et al., 2018), and Caron et al., 2018 (GIA)).

Thus, although the combined uncertainties are quite large (usually >2 mm/year) and thus conservative estimates, the global population-weighted mean VLM is still highly significant.

Please note that additional case studies matching high-resolution InSAR data with population datasets would still not be sufficient to fully assess their influence on global mean statistics, as this would require high-resolution population data worldwide. For this reason, we believe that our approach to account for spatial uncertainty in the InSAR data provides an adequate representation of this uncertainty. We'd also like to highlight that we only use population data as weightings to investigate general tendencies, but do not aim to provide a thorough risk analysis or high-resolution risk maps. Thus, we hope the reviewer agrees with our decision to maintain the focus on the role of VLM and its associated limitations and uncertainties. At the same time, we acknowledge that, for coastal risk

assessments, other sources of uncertainty may dominate. Accordingly, we added the following clarification (L636):

It should be emphasized that this analysis isolates only contributions of observational VLM uncertainties from other uncertainty sources present in digital elevation models, protection estimates, and population data, which are fundamental for coastal risk assessments [Hinkel et al., Kulp and Strauss, 2019].

Lines 152-158. I appreciate the discussion here on the limitations of statistical averages of VLM with respect to the real spatial variability within a delta. In the sentences that follow it becomes clear that the authors treat InSAR rates as a value of elevation change (VLM+sediment accretion). “It should also be noted that the reported InSAR rates usually represent total elevation change in non-urban dynamic landscapes, which is the sum of vertical land motion and vertical accretion, except for the Mississippi delta, where we account for vertical accretion using a dedicated estimate of Nienhus et al., 2020”). This conclusion is in contrast to the earlier discussed assumption that InSAR reflect VLM – here it says it does not. While the authors are aware of the considerations to be made as they do discuss it (e.g. 310-323), the implications also needs to be considered (or potential consequences at least made explicit) in the workflow decisions and result presentation.

In addition, the ability of InSAR to measure elevation change is an assumption on its own, which may very well not be the case. 1) the ability of InSAR to properly capture sediment accretion in an active sediment depositional setting is far from certain – especially when accretion rates are in the order of 1-2 mm/yr, which is lower than expected InSAR accuracy, especially in densely vegetated settings. The InSAR used (depending on type and processing of the InSAR) may/may not include vertical accretion to some extent – although to which extent remains uncertain for such large scale studies with limited validation on their effectiveness to register accumulation under different circumstances, environments and vegetation types. 2) many of the studied deltas are densely populated, which means that any human-made structure (roads, buildings, grave stones etc) acts as reflectors for the InSAR signal, and per definition do not capture sedimentation effects. As the InSAR that is used here is processed into spatial pixels (in the process merging the response of different reflectors into a single pixel). Depending on the specific local situation, the InSAR rates provide a ‘smoothed’ signal or VLM reflectors (either with or without foundations and insensitive to sedimentation) and, potentially, elevation change observations.

We are thankful for this very informative comment and agree that the current description was inconsistent, as it implied certainty that InSAR captures vertical accretion. In reality, there is uncertainty regarding the extent to which InSAR-derived rates are affected by vertical accretion. We have therefore revised the text to ensure consistency from the beginning of the manuscript:

L121: Here, our assumption is that the InSAR rates are reflecting VLM. This is supported by the estimated accuracies of the datasets from validations with GNSS measurements, which are usually between 1-2 mm/year (see also Methods, Fig. S2 and Table S1). Nonetheless, several factors and potential biases may violate this assumption, and are therefore discussed in detail in the main article.

L201: It should also be noted, that there is uncertainty regarding the extent to which InSAR-derived subsidence rates may be underestimated, either because shallow subsidence is only partially observed (when reflectors such as buildings are founded on deeper layers), or due to the influence of vertical accretion in dynamic, non-urban landscapes (Törnqvist and Blum, 2024, Minderhoud et al., 2025). that the reported InSAR rates usually represent total elevation change in non-urban dynamic landscapes, which is the sum of vertical land motion and vertical accretion. Estimates of these components are usually not available on a global scale, except for the Mississippi delta, where we account for vertical accretion using a where a dedicated estimate subsidence map based on RSETs (Rod-surface elevation tables) is available of (Nienhuis et al., 2020) that represents both shallow and deep subsidence. Therefore, in most cases, partially observed shallow subsidence and accretion are remaining sources of uncertainties these in InSAR derived estimates (see also discussion).s represent lower bounds of actual subsidence.

And in the discussion:

L356: Next to the discussed formal, spatial and cross-validation uncertainties, which are here treated as random non-biased effects. First, as mentioned above, the InSAR data used here typically represent total surface elevation change and do not distinguish between shallow and deep subsidence or vertical accretion. it remains uncertain whether InSAR measurements fully capture the combined effects of shallow and deep subsidence, or to what extent they are influenced by vertical accretion processes. These factors can be differentiated using in situ data (rod surface elevation table (RSET) and marker horizon records, e.g., Nienhuis et al., 2017, Santillan et al., 2023, or vertical well extensometers), which is done for the Mississippi Delta. However, such local information is not yet available at the scale resolved by the InSAR data, which may also contribute to differences with respect to NI21b, who also aggregated data based on such in situ information. Second, if reflectors such as buildings have different foundation depths, this can lead to high spatial variability due to differential settlement between these reflectors. Buildings with deeper foundations are also less affected by shallow subsidence. This could cause an overall underestimation of the population-weighted average subsidence, since it strongly depends on InSAR estimates from the most densely populated areas and may also contribute to differences w.r.t. NI21b [Minderhoud et al., 2025].

We hope that these changes are satisfactory, but we are happy about adding further details if required.

Line 156:, “except for the Mississippi delta, where we account for vertical accretion using a dedicated estimate of Nienhus et al., 2020. “ This is confusing and the

decision is not explained. Only when reading the methods it becomes somewhat more clear why the authors did this. This is the only delta in which InSAR was substituted with a RSET derived map (thereby excluding the vertical accretion). I suggest adding more explanation here, as this will also help to provide a better underpinning of the following sentence:

Line 157 “ Therefore, in most cases these estimates represent lower bounds of actual subsidence.” It is unclear what the InSAR data provides for such environment (see above) and requires a more thorough consideration and substantiation – as it may not be the case – e.g. InSAR in wetland areas mainly reflects human structures which may be sinking faster than the surrounding wetlands due to additional loading effects.

Line 170 “ although this cause is speculative “ I appreciate this addition, and agree with this.

We appreciate this important comment and have revised the text accordingly, as shown in the previous response. We now take a more neutral stance regarding whether InSAR provides lower-bound estimates of subsidence, while acknowledging that if shallow subsidence is only partially captured, this could have global implications.

3) Uncertainty propagation of individual dataset into the final results

The authors go to great lengths to quantify the errors for the InSAR datasets in relation to GNSS results (showing that the overall error for the different InSAR datasets floats somewhere between 1-1.5 mm/yr). In addition, the authors argue that the InSAR datasets capture a large part of the coastal population (sinking cities/deltas etc). Other datasets may have similar errors/uncertainties involved). It would be useful to add a spatially explicit worldmap in addition to the maps of fig. 2, in which this uncertainty is shown, based on the underlying dataset used.

When I look at the data provided for the deltas (delta_VLM.xlsx), the error of the InSAR with the GNSS seem to be dominantly one-directional (InSAR subsidence > GNSS subsidence). Segments on average sink 1.25 mm/yr (0.85mm/yr median) faster than the GNSS rates. While this could be due to an under registration of the true rates by GNSS, it needs to be carefully considered/discussed, also in relation to potential overestimation stemming from spatial averaging practices.

We agree; the limited discussion of uncertainties/biases was also noted by the first reviewer. As explained in the previous comments, we've added more analyses of different uncertainty types and their effects in the global estimates. We have also clarified that the estimates in the *xlsx file, specifically the GNSS rates and the averaged InSAR data are not fully comparable, because they don't represent collocated estimates (and because GPS measurements are only point measurements). Note that Ohenhen et al., 2025 provide a comparison with co-located estimates and report an RMS of 1.2 mm/year. We've added in the text L532:

Here, we report delta averages from InSAR (averaged over the entire delta surface, or averaged over the InSAR data points, that are interpolated onto the DIVA segments), as well as from GNSS point-measurements, and a combination of InSAR and GNSS. Note, that these estimates sometimes differ substantially, which is mainly due to the large spatial variability of VLM, and the heterogeneous coverage of GNSS stations in the deltas. Accordingly, these results cannot be directly compared as they do not represent co-located (InSAR-GNSS) estimates.

Inspired by the comment on the biases in the SI file, we've added some more analyses of the influence of theoretical biases in the different InSAR datasets, and their relative importances for the global mean statistics (see SI Fig. S7):

This could cause an overall underestimation of the population-weighted average subsidence, since it strongly depends on InSAR estimates from the most densely populated areas and may also contribute to differences w.r.t. NI21b [Minderhoud et al., 2025]. Expectedly, we find that the global population-weighted average subsidence (as provided in Table 1) is most sensitive to possible biases in the city and delta datasets: A hypothetical bias of 1 mm/year in either of these datasets would translate to a ~0.3 mm/year bias in the global averaged VLM.

The quantified errors/uncertainties are not reflected back in the final results and conclusions. With the above suggested error map – the authors will be able to substantiate their findings and instead of reporting a single value for VLM and amount of people experiencing this rate the uncertainty range can be provided (e.g. 307 million (+/- xxxx). For example, for Figure 2C an error envelope around the results. By providing a transparent and consistent propagation of errors/offsets in the processing framework, the results will be much more robust and future proof. In addition it will also help to pinpoint the sources of major uncertainties and set the agenda for future improvements.

As demonstrated in the previous comments, we have followed all of these suggestions and:

- Propagated the different VLM uncertainties in the global mean results (Fig 3.)
- Discussed relative contributions of data uncertainty/resampling (SI Fig. S7), and compared uncertainties across datasets (SI Fig. 6d)
- Added uncertainty envelopes to the CDF in Fig. 2c, and reported uncertainty intervals for the affected populations (the Figure is shown further below in a response to Rev3)
- Discussed influences of possible biases in InSAR products on global statistics

These new results have also implications for our conclusions (L429):

We advocate for a re-assessment of sea-level projections and impact assessment studies using a hybrid VLM data approach as presented here, incorporating all available data from different sources. Focusing future improvements on the most heavily populated and rapidly subsiding coastal regions is essential, also because VLM estimates in these areas remain among the most uncertain. This would certainly benefit from joint community efforts, which require. ~~This requires~~ the incorporation of the

effects of local VLM and realistic and consistent data uncertainty estimates, ~~with a main focus on the areas with highest coastal population.~~

4) The sea level change used in this study was inferred with a linear trend from recent observation starting from 1993. They authors argue this is only a few years before the VLM product (which one?). For sure this does not hold true for the INSAR, which is (I think all?) from the Sentinel 1 era (starting end of 2014). So there is a larger temporal discrepancy between the Sea level data and the area where InSAR is used. In addition, the authors use a linear function for sea level change, which in itself likely creates an underestimation of contemporary SLC (which is often non-linear). This approach likely underestimates contemporary sea level rise, but it is directly compared to (a.o.) InSAR-estimated VLM rates two decades younger, which likely (and unfairly) increasing the discrepancy between VLM and SLR. This approach needs to be re-evaluated, as most of the high VLM rates attributed to populations only covers the last decade – providing a better sea level change rate estimate that matches this time period will results in more fair comparison.

We appreciate this comment and agree that using absolute sea level observations over a longer period may underestimate more recent accelerated changes. We have therefore repeated the analysis comparing the trends over 1995-2020 and 2010-2020, as shown in the figure. (Note that although the data starts in 1993, we actually computed trends from 1995 in the previous version, which was corrected now).

Figure S8: Absolute sea level changes derived over different periods. Shown are (a) coastal trends [mm/year] derived over 1995-2020, over 2010-2020 (b), as well as the differences between the two periods (c).

As can be seen, computing absolute sea level trends over shorter periods of time can cause interannual/or natural climate variability to supersede the long-term climate driven trends, that are most meaningful for our analysis. In fact, over the more recent period there are much higher trends in Western Hemisphere, the Gulf of Mexico (GOM), Europe, and the Arctic Ocean, and much lower rates in Southeast Asia/Australia. Therefore, the population weighted ASL change changes from 3.15 mm/year (1995-2020) to 1.63 mm/year (2010-2020), since the more populated Asian coasts face a relatively lower ASL change over this period. This can be caused by a variety of factors, e.g., climate variability such as ENSO, or

other natural effects as demonstrated by Stammer et al., 2013, Han et al., 2019, or Dangendorf et al., 2023.

Thus, since the longer observations are more robustly reflecting the underlying long-term climate change signals, and we do not aim to provide here a full SL budget analysis accounting for the different drivers (forced/natural), we argue that using this period is the best compromise for our analysis. We still acknowledge the more recent accelerations in the rates in the text (L267):

Therefore, subsidence (with a global population-weighted average of 2.8 mm/year) currently contributes almost as much to the population-weighted RSL change as the climate-driven ASL component (3.15 mm/year). In the future, the accelerating climate-driven ASL change is widely expected to become increasingly dominant [e.g., Hamlington et al., 2024, Fox-Kemper et al., 2021], but future subsidence rates need more assessment to be sure.

(L645) The monthly altimetry absolute sea level (ASL) anomalies are used from 1993-01-01 to 2020-12-31. Note that these measurements start earlier than most of the InSAR observations. Therefore, the overall global ASL trends over this period are slightly lower than rates over more recent periods, e.g., from 2010 onwards, due to accelerations in the global mean sea level [Hamlington et al., 2024]. However, since ASL trends computed over shorter periods (e.g., from 2010) are much more strongly dominated by interannual climate variability (see SI Figure S8), leading to lower rates particularly in Southeast Asia, we still use the trends over the entire period. Note that although the time scale of the absolute sea level changes is slightly longer than that of the VLM product (which starts in 1995), this will not significantly affect the results presented here.

We hope these changes sufficiently clarify the importance of ASL changes.

Stammer D, Cazenave A, Ponte RM, Tamisiea ME. Causes for contemporary regional sea level changes. *Ann Rev Mar Sci.* 2013;5:21-46. doi: 10.1146/annurev-marine-121211-172406. Epub 2012 Sep 27. PMID: 22809188.

Han, W., Stammer, D., Thompson, P. *et al.* Impacts of Basin-Scale Climate Modes on Coastal Sea Level: a Review. *Surv Geophys* **40**, 1493–1541 (2019).
<https://doi.org/10.1007/s10712-019-09562-8>

Dangendorf, S., Hendricks, N., Sun, Q. *et al.* Acceleration of U.S. Southeast and Gulf coast sea-level rise amplified by internal climate variability. *Nat Commun* **14**, 1935 (2023).
<https://doi.org/10.1038/s41467-023-37649-9>

Hamlington, B.D., Bellas-Manley, A., Willis, J.K. *et al.* The rate of global sea level rise doubled during the past three decades. *Commun Earth Environ* **5**, 601 (2024).
<https://doi.org/10.1038/s43247-024-01761-5>

Minor issues:

Line 80: please clarify the 2D displacement mentioned here, this is confusing and it depends on how the InSAR is processed and whether or not combined with ascending/descending tracks – in principle providing a 3D observation of the movement (vertical, east-west, north-south). Perhaps here the authors have a specific InSAR processed product in mind.

We have corrected and clarified this point. Our intention was to emphasize that the processed InSAR products provide spatially, sometimes almost continuous, two-dimensional maps (lon/lat) of vertical land-motion estimates, in contrast to the spatially sparse, point-based GNSS observations (L81):

While these techniques represent sparse point-wise measurements, InSAR ~~provides 2D~~ can provide velocity estimates in satellite's line of sight for a large number of coherent reflectors, such as buildings or infrastructure. Using known ground velocities, typically derived from GNSS measurements referenced to a global reference frame, these line-of-sight velocities can be transformed into displacement ~~two-dimensional VLM~~ estimates at a resolution of several meters ...

Fig. 2c. discrepancy between caption and text in figure. 1-2-3 mm/yr in caption, in figure 0-1-2 mm/yr.

Corrected

Figure S4/5. The statistical information related to the error bar (black lines) is missing.

We have added this information to the Figures caption.

Line 146 (and elsewhere): typo in Nienhuis reference.

Corrected

Fig. 6D: Pop. Below 10m, add the 1e8 in this line to make it consistent with rest of the fig. – it is now hidden.

We changed this figure, only showing uncertainties now and updated it with the new information from New Zealand, and global deltas (which were missing in the previous version).

PM

Reviewer #3 (Remarks to the Author):

The paper presents a synthesis of globally available constraints on vertical land motion and sea surface elevation to provide a population-weighted measure of impact from relative sea level rise, focusing especially on major cities. The contributing datasets include InSAR, GNSS, tide gauges, altimetry and GIA models and sea surface satellite altimetry. The focus is on how the imaged relative sea level change impacts the human population. It will probably get some attention since sea level rise is of such high concern among the slow environmental risks facing society today. The paper is generally well written and seems to fit the appearance of articles published in Nature Communications in terms of length, voice, graphic quality, etc.

Overall it will be a nice contribution to the journal. The title is catchy, and it is interesting that the coastal human population tends to be concentrated in areas where VLM sign is on balance subsidence and are low lying. While this has been known for quite a while, the analysis quantifies the impact, sorts cities based on their exposure to subsidence, and draws out some interesting pieces of information about how sensitive the conclusions are to the constraints brought by different datasets (e.g., InSAR, GNSS, GIA models).

The dependence that InSAR has on GNSS is probably understated in the text. All InSAR analyses that pertain to vertical land motion of significance to sea level rise need to be aligned to geocentric reference frame in order to be compared to sea level trends. This is available only in the GNSS, and the InSAR relies on it completely. It was not immediately clear to me whether the difference InSAR data sources were aligned separately into the same GNSS reference frame. They appropriately showed level of agreement between GNSS and InSAR for each dataset in Table S1. The mentioned transforming the InSAR from ECMS with a latitude-dependent function, but did they do that for the other datasets too?

Thank you very much for your comments and your positive review! We fully agree that the global GNSS estimates are fundamental for consistent InSAR VLM estimates. Thus we added in L83:

Using known ground velocities, typically derived from GNSS measurements referenced to a global reference frame, these line-of-sight velocities can be transformed into displacement two-dimensional VLM estimates at a resolution of several meters—and is. Accordingly, InSAR is being increasingly used in many large coastal cities or more widely _[Raucoules et al., 2013, Crosetto et al., 2020, Buzzanga et al., 2020, Naish et al., 2022, Ohenhen et al., 2023, Ao et al., 2024, Thiéblemont et al., 2024], but it remains strongly dependent on the quality and availability of underlying GNSS estimates [Blewitt et al., 2018, Hammond et al., 2018]-

We added another section in the discussion (L372):

Such remaining uncertainties underline the importance of dense, continuous global GNSS networks (e.g., as provided by the Nevada Geodetic Laboratory, Blewitt et al., 2018). These measurements are the prerequisite to derive comparable InSAR VLM estimates in a global reference frame and are thus fundamental to reduce biases in regional estimates. Yet many of the world's most populated cities and deltas - particularly in South, Southeast, and East Asia, as well as major African coastal centers - still lack the observational coverage available in Europe or the United States (e.g., Ohenhen et al., 2025; SI Fig. S6a–S6b), further emphasizing the need for continuous and global observations (Blewitt et al., 2018). Comparable to limitations in InSAR estimates, GNSS reference stations with deep anchor depths may overlook shallow compaction in landscapes where such processes are significant, leading to underestimated total subsidence. Therefore, given the key role of deltas and cities in controlling the rates of sea-level rise felt by the average person, these factors must be considered, and InSAR data should ideally be analyzed in synergy with in-situ (e.g., RSETs and GNSS) observations.

We also clarified that all observations are separately aligned to ITRF2014, which is the reference frame of the GNSS stations to which the InSAR datasets are aligned to (by the creators of these datasets). Among the datasets, EGMS was the only one provided in a different reference frame (ETRF2000); therefore, we applied this simple transfer function exclusively to that dataset:

L440: All datasets (GNSS, altimetry and InSAR) are referenced to the ITRF2014 [Altamimi et al. 2016], except for EGMS, which is transformed into ITRF2014 using an empirical transfer function.

Please note, that in addition to the more detailed responses below, the main change in the updated manuscript is the improved discussion and treatment of uncertainties in the InSAR data. Now, we integrate formal, spatial and cross-validation uncertainties and also discuss hypothetical biases. We have also integrated another InSAR dataset for New Zealand. Accordingly, we updated most of the Figures.

I make some more specific comments and suggestions below.

Detailed Comments

1. Not sure the phrase "climate-driven" needs to be in the title. Aren't all changes in sea level climate-driven since the level of the sea is a part of the whole climate system?

We dropped climate-driven from the title.

59. "Note that there are little or no human-induced uplift processes, so the effect of

human-induced VLM acts overwhelmingly in one direction – subsidence”. While this statement is technically true, I can think of one. The California Central Valley subsidence is implicated in driving hydrological mass loss that helps create unload and uplift of the Sierra Nevada (Amos et al., 2014, Nature). But this effect is very slow, and the word “overwhelmingly” may appropriately capture the balance between anthropogenic and natural uplift.

This is a very interesting comment/paper. We added this paper writing L61 ‘... with some exceptions [e.g., Amos et al., 2024]’

83-94. Very long sentence. Could it be divided?

We changed this to: While these techniques represent sparse point-wise measurements, InSAR ~~provides 2D~~ can provide velocity estimates in satellite’s line of sight for a large number of coherent reflectors, such as buildings or infrastructure. Using known ground velocities, typically derived from GNSS measurements referenced to a global reference frame, these line-of-sight velocities can be transformed into displacement ~~two-dimensional VLM~~ estimates at a resolution of several meters ~~and is~~. Accordingly, InSAR is being increasingly used in many large coastal cities or more widely [Raucoules et al., 2013, Crosetto et al., 2020, Buzzanga et al., 2020, Naish et al., 2022, Ohenhen et al., 2023, Ao et al., 2024, Thiéblemont et al., 2024], but it remains strongly dependent on the quality and availability of underlying GNSS estimates [Blewitt et al., 2018, Hammond et al., 2018].

89. "in South, and Southeast, and East” would be better to say “in South, Southeast, and East” (link on line 307)

We corrected that.

126-127. "fact that InSAR only covers about 17% of the global coastline by length, but almost 55% of the coastal population underlines its utility to resolve VLM for human and socio-economic analyses.” It should be made clear to what extent this is because of data not being present (i.e., not covered by the satellite), versus data not being usable (e.g., decorrelation or other problem) versus data not processed/analyzed in the right way yet. Is it a matter of just more work needs to be done or is there a fundamental limitation to acquiring these data?

This is mainly due to the current availability based on the regional focus of the different studies, possibly also explaining the focus on the industrialized, most populated, and most vulnerable locations. So it’s really reflecting the current progress (note that essentially all of the InSAR datasets used here were released just within the last 5 years). We’ve added (L129):

Thanks to the abundance of recently published InSAR datasets, almost ~~55~~65% of the global coastal population is now covered by accessible measurements (see Fig. 1). Regions where InSAR VLM

estimates have been processed is available (highlighted by black-outlined markers) cover almost the entire US coast [Ohenhen et al., 2024], Europe (EGMS), ...

line 158. What is 'actual subsidence'? This phrase could be replaced with something more specific, such as 'vertical motion of solid earth'

We changed most of the paragraph where this statement appeared, now it reads (L207):
Therefore, partially observed shallow subsidence and accretion are remaining sources of uncertainties in InSAR derived estimates (see also discussion).

272-273. This is a very long section header and should be shortened.

We agree and changed it to: ~~Towards community-based efforts to overcome limitations in current VLM observations and our process understanding and future projections~~ Toward community-driven advances in VLM observations, process understanding, and projections.

278-280. Getting repetitive.

We deleted the associated sentence.

314-315. Or vertical well extensometers.

We added that.

321-323. The dependence that InSAR has on GNSS is probably understated here. All InSAR analyses that pertain to vertical land motion of significance to sea level rise need to be aligned to geocentric reference frame in order to be compared to sea level trends. This does not come across very well in the paper as it is now presented but should be stated explicitly here or in the Methods section.

We agree. As mentioned in the previous comment we highlight the dependence of InSAR in the introduction (but it remains strongly dependent on the quality and availability of underlying GNSS estimates [Blewitt et al., 2018, Hammond et al., 2018]-) and added a dedicated paragraph in the discussion (Such remaining uncertainties underline the importance of dense, continuous global GNSS networks (e.g., as provided by the Nevada Geodetic Laboratory, Blewitt et al., 2018). These measurements are the prerequisite to derive comparable InSAR VLM estimates in a global reference frame and are thus fundamental to reduce biases in regional estimates. ...)

363. "In conclusion, we advocate for a re-assessment of sea-level projections and impact assessment studies using a hybrid VLM data approach as presented here, incorporating all available data from different sources." Good statement but is not really the advertised conclusion of the paper... seems like the main point of the

analysis was to emphasize the impact of VLM on human populations. The advocacy should come after the scientific conclusion, maybe put that sentence at the end of this paragraph.

We are thankful for this suggestion; we have changed it to:

In conclusion, ~~we~~subsidence causes about half of the RSL change that is currently experienced by coastal populations. Our findings reinforce earlier work highlighting the large potential to reduce RSL change by mitigation of human-induced VLM. We advocate for a re-assessment of sea-level projections and impact assessment studies using a hybrid VLM data approach as presented here, incorporating all available data from different sources. Focusing future improvements on the most heavily populated and rapidly subsiding coastal regions is essential, also because VLM estimates in these areas remain among the most uncertain. This would certainly benefit from joint community efforts, ~~which require.~~ ~~This requires~~ the incorporation of the effects of local VLM and realistic and consistent data uncertainty estimates, ~~with a main focus on the areas with highest coastal population.~~ ~~Since subsidence causes about half of the RSL change that is currently experienced by coastal populations, this study reinforces earlier conclusions about the large potential to reduce RSL change by mitigation of human-induced VLM.~~

376-379. This will make GNSS from NGL the underlying connection to the reference frame. Please state which reference frame the GNSS is in (e.g., ITRF14). NGL recently reprocessed all data in IGS2020 so either are available, and it is necessary to know which frame is used here.

Yes, this information was missing, we added: All datasets (GNSS, altimetry and InSAR) are referenced to the ITRF2014 (Altamimi et al. 2016), except for EGMS, which is transformed into ITRF2014 using an empirical transfer function.

422. “Blewitt et al., 2019” not in reference list. Probably what was meant was “Blewitt, G., W.C. Hammond, C. Kreemer, 2018, Harnessing the GPS Data Explosion for Interdisciplinary Science, Eos, 99, <https://doi.org/10.1029/2018EO104623>” ?

Yes, we changed all references from Blewitt et al., 2016/2019 to Blewitt et al., 2018.

493. I clicked on the link to CMEMS and it made it to <https://data.marine.copernicus> but said the link is broken. Is there a more permanent link to cite?

We corrected the link:

https://data.marine.copernicus.eu/product/SEALEVEL_GLO_PHY_L4_MY_008_047/description

(The link likely contained an unintended space caused by a line break.)

559. Capitalization in references needs attention, many things that should be capitalized are not.

Thank you for this observation, we corrected the references accordingly.

Figures

Figure 1. Move legends to side of map (not on top of) so as not to interfere with seeing some places in east and west Pacific

We've changed this Figure according to the suggestions and added uncertainties as well.

Not enough contrast between areas with 'black outlined markers' and other areas, e.g., most of major land masses seem to have a black outline around the colored symbol. Could that be made gray or some other color?

As shown in the Figure, we have increased the overall size of the map and improved the contrast between the black and grey outlined markers.

Figure 2. Figure 2C indicates what fraction of the population experience a given amount of VLM or less. It would seem more informative to flip the axis so the number represented was the fraction of the population that experiences that amount of subsidence or more.

We followed this suggestion and inverted the y-axis to show how many people experience at least x mm/y in VLM/subsidence.

Figure 2D colors an entire country based on what its coast is doing. While there may be policy consistency within a given country it does lead to false impressions of national importance of sea level rise in some cases. For examples, the Congo in Africa is virtually land locked, it only has ~50 km of coastline, so its exposure to risk may not be proportional to the area colored red. The US had a constant RSLC on the map whereas in reality its risk profile to RSLC is highly spatially variable.

Yes, we agree that the country-aggregation scale is sometimes problematic to show coastal variables/statistics. Also, the current color scale doesn't really differentiate between the different population sizes of the countries. That's why we suggest to use a 2D colorscale: Red-pink-Blue: Higher-smaller population weighted RSLC, transparency: high to low: Larger to smaller overall coastal population size:

Figure 2. Contribution of VLM to relative sea-level change. **Aa)** RSL change as the combination of ASL change (from CMEMS over 1995-2019) and VLM (based on the VLM reconstruction OE24), GIA [Caron et al. 2018], where

OE24 has missing data) and InSAR from EGMS, Ohenhen et al., 2024, 2025, Shirzaei et al., 2024, and Ao et al., 2024. **Bb)** shows the number of people living below 10 m elevation on a logarithmic scale. The size of the data points are non-linearly scaled by their value. **Cc)** Cumulative distribution of the contribution of coastal length-weighted and population-weighted VLM to RSLC (positive sign = subsidence). The inverted cumulative frequency on the y-axis thus refers here to the share of population experiencing at least the subsidence rate shown on the x-axis. Solid lines represent the hybrid VLM estimates of this study, the dashed line represents the same dataset without InSAR, and the dotted line represents the total averaged VLM estimate from NI21b. The shaded region surrounding the solid red line (i.e., the VLM estimate) denotes the 95% confidence interval. This uncertainty is estimated from a bootstrapped distribution of CDFs generated by perturbing the VLM rates using normally distributed random errors derived from the estimated uncertainties (see Methods). We also show the fraction of people who experience subsidence rates ~~of at least one, two or three~~ of >0 , >1 , and >2 mm/year (using estimates from the hybrid reconstruction). **Dd)** shows both, the averaged, population-weighted RSL change per country (as indicated by the colors ranging from blue to red, i.e., low to high RSL change), as well as the total coastal population, which is displayed by the modulation in the transparency of the colors. That is, it shows the RSL rates that are on average experienced by the LECZ population in each country.

We also mentioned that the country-aggregated statistics may not resolve existing within-country differences (L232):

Clearly, these country-averaged values strongly depend on the sub-national distribution of population centers and VLM, and there is often substantial variability within countries. As an example, the population-weighted standard deviation of RSL change can be as large as 7-9 mm/year for countries like China or Indonesia (see SI Fig. S4), and differences between the US Gulf and West coast (see also Fig. 1.a) are not resolved at the country-aggregation scale.

Figure S2. What is the third row? very narrow, hard to see what this is. Does it add any insight or interpretive value to the data?

This row shows boxplot of differences. We believe it is helpful as it can show any potential biases of individual datasets. We added the associated description.

What is the value of “e” in the equation in the caption? The Thielblemont et al., 2024 citation is listed as in review.

Thank you for noting this, there was a small typo (it should be $2e-4$), which is corrected now:

$$\Delta V_{\text{fit}} = -2 \times 10^{-4} \phi^2 + 0.04\phi - 0.85$$

This equation with latitude dependence is a very long wavelength correction for the difference between InSAR and GPS. Consequently, errors in the InSAR will still be present in shorter length scales (e.g. on the scale of a single swath of the satellite

pass). Its possible that alignments are done within the InSAR products that are used as input to this analysis. If so please describe those details in the Methods section.

Yes, there can definitely be still errors in the InSAR data at small wavelengths. Note, that we used this transfer function solely to align the different reference frames of the EGMS product and the other datasets: We write in the Methods (L474):

The EGMS VLM data were referenced to the European GNSS stations in the European Terrestrial Reference Frame ETRF2000. To reference these velocities to the International Terrestrial Reference Frame (ITRF2014, Altamimi et al. 2016) and to ensure a consistent reference frame for the entire hybrid dataset, we apply a latitude (ϕ)-dependent transfer function presented by Thiéblemont et al. (2024): $\Delta V_{\text{fit}} = -2 \times 10^{-4} \phi^2 + 0.04 \phi - 0.85$. This second-order polynomial transfer function (see also SI Fig. S2) is based on the differences between VLM from GNSS in the ETRF2000 and ITRF2014 reference frame.

We hope that this clarifies this question.

Figure S4. It looks like the red and blue colors grade towards white but there is no color bar to say what the different shades mean.

Yes, these colors are scaled by the amplitude of the pop.-weighted RSLC. We added the associated colorbar.

Figure S6B. What does the dot color mean? Horizontal axis needs annotation/labeling (caption says its population).

We improved this Figure including labels, description of colors and updated uncertainty estimates.

Reviewer comments

Reviewer comments are shown in **bold**, author responses in regular font, and changes to the manuscript are highlighted in red. All new line numbers refer to the final version (without changes).

Reviewer #1 (Remarks to the Author):

I think the comments were carefully and adequately considered by the authors and I recommend the manuscript for publication.

Dear Reviewer,

Thank you very much for your constructive comments and efforts!

Best,

Julius Oelsmann

Reviewer #3 (Remarks to the Author):

The authors addressed my comments, I have only one remaining suggestion:

In Figure S8 there are three panels. First is absolute sea level change rate for the full period (a), second is ASLC rate since 2010 (b). The third is ASLC rate full period minus ASLC rate since 2010 (c). This results in areas that have accelerated ASLC rate appear negative in the third panel. Wouldn't you want it the other way around so it appears that positive changes in ASLC rate appear positively in the third panel? That seems to make more sense to me since a positive value suggests a positive acceleration in sea level rise.

Dear Reviewer,

Thank you very much for your review! We also agree that it makes sense to change Figure S8, as suggested. We have updated this Figure accordingly.

Best,

Julius Oelsmann

Reviewer #4 (Remarks to the Author):

The authors produced an interesting synthesis of available vertical land motion data along populated coasts to be used for relative sea-level rise assessments. I think this is a very useful endeavor. I think what they produced is a critical step forwards, although like the authors mention, a VLM product should consist of more than only observations, it should also consist of subsurface information and drivers of subsidence. But, with respect to the large spatial scale of this study, I understand that cannot be achieved at this stage.

Dear Kay Koster,

Thank you very much for your constructive and comprehensive review. We believe that your comments have substantially improved the manuscript. In particular, your suggestions to streamline Section 1 and more clearly distinguish between the new results of this study, background knowledge from the literature, and discussion points were extremely valuable, and we have revised the manuscript accordingly. As explained below, we have also clarified why we use Nicholls et al. (2021) as a benchmark to motivate our analysis and to place our results in the context of previous global assessments. Finally, we now describe the limitations of our approach more clearly, especially the need for additional efforts to actually understand the underlying subsurface processes that cannot be resolved from surface observations alone. We also thank you for your many other detailed comments, which we have carefully considered throughout the revision. We hope that our revisions in response to these comments to address your concerns satisfactorily.

Best,

Julius Oelsmann

Throughout the manuscript I made some comments which can be easily addressed. Some major points:

- Please already mention in the Introduction that a VLM product based on observations is only half the story, the other half, which you don't use in the analysis, regards the use of subsurface data, subsurface use, policy etc. This aspect is now only briefly mentioned in the Discussion, but that is too late. We already know for years that VLM cannot be understood using geodesy alone, so please follow the state-of-the-art; there are numerous relevant studies to be found.

Thank you for this comment, we have clarified this now in the introduction as well:

L112: Therefore, we take a pragmatic hybrid approach and compile the most comprehensive VLM dataset from the available sources, revealing small scale VLM including the most populated coastal metropolitan areas and analyze current changes of RSL rise (see Materials and Methods). *While these observational estimates provide important constraints on present-day RSL and for coastal risk assessments, we recognize that these data sources capture only the surface expression of VLM. Since VLM is driven by a variety of subsurface processes, which are influenced by geological conditions, resource use and subsurface management, a full understanding and projection of VLM thus requires integration of geodetic observations with subsurface data and process-based models [e.g., Koster et al., 2018; Minderhoud et al., 2020; Shirzaei et al., 2021].*

- Second, I read NI21b a while ago, but I can assure you, I don't know its details anymore. So please, give a brief summary in the beginning. That said, I am not entirely convinced that you should compare this study to NI21b at all. It looks like you are comparing apples with oranges (observations vs. expert judgement – why would you do that?). I also see that some of the NI21b authors are involved in this new research, so why the obsession of comparing yourself?

We are thankful for this comment. We agree that the current explanation of their approach, as well as the relevance of comparing our results with their earlier work should be better explained. Thus, we first provided a better overview over their approach and existing limitations (introduction):

L69: Based on such estimates, Nicholls et al. [2021b] (hereafter NI21b) highlighted that the coastal population (*living below 10 m above sea level*) is preferentially concentrated in subsiding areas, especially in susceptible cities and deltas. They estimated that the global coastal-population weighted RSL rise (7.8 - 9.9 mm/year) strongly exceeds the global coastal average (2.6 mm/year), *establishing this study as an important benchmark that demonstrated the global-scale significance of subsidence for coastal residents*. However, one of the limitations of NI21b is that no direct geodetic VLM measurements (e.g., by InSAR (Interferometric Synthetic Aperture Radar) or GNSS (Global Navigation Satellite Systems)) were considered. *Instead, subsidence rates were*

compiled from numerous individual studies and meta-analyses (e.g., Ericson et al. 2006), which rely on a heterogeneous set of approaches, including borehole extensometers, leveling surveys, groundwater-extraction-based models, etc., and simplified assumptions. Thus, additional limitations of these estimates include their low spatial resolution, particularly in large deltas and cities, where they are often represented by single values averaged over entire regions, thereby neglecting small-scale variability, as well as their frequent reliance on expert judgement, especially in deltas where observational constraints were sparse. Accordingly, the impact of VLM - as determined by currently available high-resolution observing systems - on contemporary RSL change and human exposure to sea-level rise is unclear.

We make this comparison because NI21b is, to our knowledge, the most influential previous global assessment of the contribution of subsidence to population-experienced RSL change, and therefore provides the key benchmark and motivation for the present study. Our intention is not to compare observations with “opinions”, but rather to show transparently how a new observation-based global synthesis changes, refines, or confirms conclusions from the previous state of knowledge. We believe this comparison is important for placing our results in context and for demonstrating the added value of direct geodetic observations relative to earlier global assessments.

• I think making a global coastal VLM map is a proper scientific product. It’s a product most people can understand. I cannot follow the subsequent steps in which you make a derivative of population-weighted VLM. I have no clue what it physically means or why it is relevant at all. Please reconsider that section or maybe shorten it and place it in the Discussion.

Thank you for this comment. In response, we have clarified more explicitly why we perform the population-weighted analysis. As explained further in our responses to the detailed comments below, population-weighted averages of VLM and RSL change are meaningful because they represent the conditions experienced by the average coastal resident, rather than the average coastal area. They show that people tend to be located in subsiding coastal areas and this exacerbates the human implications of subsidence. This is particularly relevant at the global scale, where subsidence and coastal populations are highly unevenly distributed and often intersect in densely populated low-lying regions.

The population-weighted statistics therefore provide an indication of the importance of coastal subsidence for risk to people, relative to current climate-driven absolute sea-level change. They also allow us to assess how observational uncertainties and data gaps affect our understanding of the regions where high subsidence rates coincide with high population density, and thereby help identify priorities for future work on process understanding, observational coverage, and data integration. Addressing these questions is a central motivation of the study and directly follows from our research questions: L123: (1) What is the current (1995-2021) rate of RSL rise experienced by coastal populations? (2) How much of these changes is driven by ASL rise and how much by VLM?

We also added a couple of explanations to clarify our approach:

L71: 'They estimated that the global coastal-population weighted RSL rise (7.8 - 9.9 mm/year) strongly exceeds the global coastal average (2.6 mm/year), **establishing this study as an important benchmark that demonstrated the global-scale significance of subsidence.** However, one ... '

Section 2, L226: 'We follow NI21b and consider the global population-weighted mean estimates of averaged RSL change to understand what the average coastal resident experiences, as opposed to what the average coastal area experiences, which is how sea-level data is normally weighted. **This distinction is important because coastal populations are highly unevenly distributed, with large numbers of people concentrated in low-lying urban and deltaic regions where RSL change can differ substantially from the mean (weighted by coastal length). Population weighting therefore provides a more appropriate measure for assessing human exposure to RSL change and the associated contribution of subsidence to RSL hazard on a global scale. By contrast, length-weighted estimates may be more relevant for applications focused on coastal land changes.**

• Large parts of the Results section aren't scientific results, please check that and transfer sections to Discussion/Introduction.

Thank you for your careful comments here! There are indeed some parts that should be moved to the discussion, or are rather descriptive and based on previous knowledge. We cleaned the result section according to your suggestions, and clarified which of the results are novel to this work.

Detailed comments:

L39: "susceptible areas" -- pls mention some concrete examples where VLM outpaces ASL, preferably from different continents to stress it is a global phenomenon

We added: 'like global deltas and coastal cities located on deltas', and also added Ohenhen et al., 2026 (on river deltas). Note that more specific examples follow.

L40: VLM estimates in general are fine, maybe not globally in a single model/mapping product, but local studies are sound - although such an analysis consumes various data types (observations, subsurface use/properties etc.). It is the projection of VLM that is tricky, not even globally, but locally we can hardly manage that.

Thank you for this comment, we changed it to 'projections'

L45: spatially I assume, not temporal

Correct, we added 'spatial'

L46: Observations are only a part of understanding VLM, as observations are merely a symptom of VLM. The true VLM processes occur below the surface. You need to stress this, because without information of the subsurface, the subsurface use, and how the subsurface response to this in terms of VLM, you cannot project anything. For a land subsidence expert like myself, I don't believe any projection based on observations alone, you need more data types. (you refer here to the sentence: As many **spatial** observational gaps are being filled with new data [Thiéblemont et al., 2024, Ohenhen et al., 2024, Shirzaei et al., 2024, Ao et al., 2024, Ohenhen et al., 20256], we seek to synergize globally available VLM data to better understand its influence on relative sea-level (RSL) estimations especially for densely populated coasts where the implications are greatest.)

This is related to the next comment as well:

47: you need to outline why you think it is enough to only use observation for this - i understand you cannot gather subsurface data etc of all the global coastlines, but this needs to be mentioned somewhere. See also my comments above, they are a bit of the same nature

We agree that this has to be clarified! As a response, we added further below in the introduction the caveats and limitations that come along subsurface processes:

L112: Therefore, we take a pragmatic hybrid approach and compile the most comprehensive VLM dataset from the available sources, revealing small scale VLM including the most populated coastal metropolitan areas and analyze current changes of RSL rise (see Materials and Methods). **While these observational estimates provide important constraints on present-day RSL and for coastal risk assessments, we recognize that these data sources capture only the surface expression of VLM. Since VLM is driven by a variety of subsurface processes, which are influenced by geological conditions, resource use and subsurface management, a full understanding and projection of VLM thus requires integration of geodetic observations with subsurface data and process-based models [e.g., Koster et al., 2018; Minderhoud et al., 2020; Shirzaei et al., 2021].**

Koster K., Stafleu J., & Stouthamer E. (2018). Differential subsidence in the urbanised coastal-deltaic plain of the Netherlands. *Netherlands Journal of Geosciences*, 97, 215-227. <https://doi.org/10.1017/njg.2018.11>

L48: VLM in general, not only coastal VLM. Coastal VLM is not a thing or process on its own, it is just VLM which happens to be near shore. An exploited gas field, aquifer, GIA, or coal seam for example can be anywhere. Of course, soft plains can often be found, but not limited to, near coastal areas.

We deleted 'coastal'

L52: erosion is normally not regarded as a VLM process

We deleted 'erosion'

L52: the study area of Törnqvist is not near coastal btw, but inland outside the M-delta

Would you agree with keeping it, as it may be representative for large coastal deltas?
We also added the following citation now.

Zoccarato, C., Minderhoud, P.S.J. & Teatini, P. The role of sedimentation and natural compaction in a prograding delta: insights from the mega Mekong delta, Vietnam. *Sci Rep* 8, 11437 (2018). <https://doi.org/10.1038/s41598-018-29734-7>

L53: Humans are the main driver of present day VLM. I would mention those first, and then go to the natural processes of VLM, which often prevail at longer timescales. For the general subsidence/VLM expert working on present day challenges, GIA is just a nuisance (except when you work near the center of the former ice sheets). When working on VLM rates during lets say the past years/decades, human induced processes are more relevant. Since you are not focusing on Holocene timescales, you should mention the human induced drivers first. Maybe also mention were the different processes matter (GIA up north e.g.)

This is a great suggestion, we changed that accordingly.

L59: i wouldnt say alluvial plain, i would mention something 'tidally'. Most soft coastal plains are formed by tidal processes, not rivers

Changed to 'or cities built on low-lying deltaic and coastal plains such as Bangkok'

L57: the Mekong delta did not subside with 10 cm/yr I think, rather 10 cm in a decade. Check this pls

Thank you for noting this, we changed it to 10 mm/year.

L57: maybe mention the dominant VLM driving process as well to give the reader an idea?

We added: **Largely groundwater-extraction-related** subsidence ...

L71: So what did they use? How can you derive VLM rates without a measurement? Can you elaborate this - for the reader not familiar with that paper this statement is a bit confusing - did they use a modeling approach?

We added (see also response to major comment): They estimated that the global coastal-population weighted RSL rise (7.8 - 9.9 mm/year) strongly exceeds the global coastal average (2.6 mm/year)., **establishing this study as an important benchmark that first demonstrated the global-scale significance of subsidence.** However, one of the limitations of NI21b is that no direct geodetic VLM measurements (e.g., by InSAR (Interferometric Synthetic Aperture Radar) or GNSS (Global Navigation Satellite Systems)) were considered. **Instead, subsidence rates were compiled from numerous individual studies and meta-analyses (e.g., Ericson et al. ,2006), which rely on a heterogeneous set of approaches, including borehole extensometers, leveling surveys, groundwater-extraction-based models, etc., or simplified assumptions. Thus, additional limitations of these estimates include their low spatial resolution, particularly in large deltas and cities, where they are often represented by single values averaged over entire regions, thereby neglecting small-scale variability, as well as their frequent reliance on expert judgement, especially in deltas where observational constraints were sparse.** Accordingly, the impact of VLM - as determined by currently available **high-resolution** observing systems - on contemporary RSL change and human exposure to sea-level rise is unclear.

Ericson, J. P., Vörösmarty, C. J., Dingman, S. L., Ward, L. G. & Meybeck, M. Effective sea-level rise and deltas: causes of change and human dimension implications. *Glob. Planet. Change* 50, 63–82 (2006).

L75: but only on single sites

We mention this limitation later: 'While these techniques represent sparse point-wise measurements, '. We also added this limitation in the discussion.

L95: just for the record, projecting VLM is extremely difficult and super local. The main unknown for projecting VLM is human decision. In order to know what VLM are going to be in the future, you need to 'project' policy with respect to e.g. resource extraction, drainage, urbanization etc. Keep this in mind, VLM projection is a different ball game than sea level rise projection which can rely on e.g. global climate models.

We totally agree, thank you for this comment! We hope that our combined dataset can be a useful validation constraint for any studies that, e.g., aim to link processes to VLM observations, e.g., resource usage, to the extent that these processes are well represented by our observations.

L99: what is a small scale according to you? Small scale for me is below 10x 10m, but we all have our own definition for this

Here we usually consider small scale several hundreds of m. We added that.

L103: it is still not made clear what the NI21b approach beholds

We hope that this is clearer now with the previous description. We added further information in the discussion, as well (see comment further below).

L104: current as in 2025, or 2020-2025, or 2000 - 2025 etc. What is 'current' in this context?

We define current as 1995-2021, and added it accordingly.

L105: maybe earlier you can mention how you define coastal population or coastal area for that matter.

We define that now earlier in the introduction (i.e., population below 10m).

"Results" -- The Results section is full of sections that clearly aren't 'results'. Please check this, see my comments

Thank you for this important observation, we followed your suggestions here (see other comments below).

L119: do these studies overlap temporally, or are you going to compare different years with each other. You should mention the years you are analyzing - this matters from a VLM point of view, because different years have differences in subsidence drivers/intensity

While we discuss these issues also later in the manuscript, we agree that it makes sense to clarify that right from the start. We added: Here, **we rely on several assumptions: First, we assume that all reported InSAR rates are reflecting VLM. Our second assumption is that trends from all data sources are representative of the changes over the entire period (1995-2021) considered here. This is a simplification, as InSAR based VLM rates are generally derived from relatively short and variable observation periods (~5-15 years, see SI Table S1). Uncertainties arising from potentially unobserved non-linear changes, as well as from partially observed shallow subsidence and vertical accretion, are therefore discussed further in the Discussion.**

L121: but you know that isnt the full picture. Insar doesnt say much outside built-up areas for instance, but maybe you can bypass that by the fact you focus mainly on cities

Thank you for this suggestion, we mention this limitation in L144: **First, we assume that all reported is that the InSAR rates are reflecting VLM. While this is supported by the estimated accuracies of the datasets from validations with GNSS measurements, which are usually between 1-2 mm/year (see also Methods, Fig. S2 and Table S1), several factors may violate this assumption. In particular, there is uncertainty regarding the extent to which InSAR-derived subsidence rates may be underestimated or misinterpreted, either because shallow subsidence is only partially observed (when reflectors such as buildings are founded on deeper layers), or due to the influence of vertical accretion in dynamic, non-urban landscapes [Törnqvist and Blum, 2024,**

Minderhoud et al., 2025]. Estimates of these components are usually not available on a global scale.

Note, that additional information was added to the Discussion (see comment further below).

L132: large parts of Europe, not all countries are in the Copernicus program

We changed that accordingly.

L134: are you only looking at deltas, or also coastal plains? I think you mean coastal plains, but you confuse them for deltas. In NW-Europe, there isn't any delta for instance, they are estuaries/tidal inlets in coastal plains. Pls check this and maybe rephrase this

Here, we referred to Ohenhen et al., 2026, which provided InSAR data for large deltas, we added that information, and also changed it to delta/estuaries.

L140: i don't understand this – isn't for instance sentinel1 not available globally? Or aren't there any gnss stations available to process it for all coast lines?

Yes, it could theoretically be processed everywhere, but currently not all coastal regions are covered by InSAR VLM estimates yet. We also mentioned that in L129-L130.

*The following comments largely refer to section 1 (**Global hybrid vertical land motion estimates**) and suggest that some parts should be either moved to the discussion or removed, as they summarize previous results. We believe these are really valuable suggestions, as it is currently indeed very difficult to follow what's a new result, or what's a priori knowledge. We have rearranged this section accordingly, keeping only the results that are unique to this study.*

Note, that we've kept key statistics about cities and deltas, as they are really important details of the description of the arguably most important figure (Fig 1) of this work, and also clarified better which aspects are new to this study. These aspects include: (1) in particular the city data [Shirzaei et al., 2024], which hasn't been described yet at this detail-level (this was only a data publication by our coauthor, M. Shirzaei). We also added some new estimates in deltas beyond what's covered by Ohenhen et al., 2026 (e.g., Nienhuis et al., 2020, and GNSS estimates) (2) city and delta data represent here the values aggregated on the coastal DIVA grid and include new uncertainty estimates, which are so far not published. We also added new comparative analyses of how well cities and deltas are covered with previous geodetic estimates (GNSS, interpolated data, see SI Figures S3 and S4, and SI Data File on delta statistics). (4) Information of city and delta subsidence estimates is later very important for the RSL analyses, and the comparison with Nicholls et al., 2021., who use the same DIVA grid.

Note that we have still drastically shortened some interpretations and well known facts about GIA, the US-East coast, the Netherlands and Po Plain, etc. Please find a detailed history of changes below.

L143: We know for centuries the coastal plain of the Netherlands is subsiding by human-activities - it is not a result of your research, it is a priori knowledge. It is neither a result of Thieblemont et al., The same accounts for all/most of the other areas you mention. The results section should focus on results only - you are here taking geodetic info from other studies and present them as your own, right? I think you can remove this entire section (yellow) and focus on the integration with ASL data. That is the novelty of your study

We fully agree, and removed most of these parts.

Note that we have kept: The global hybrid VLM estimate demonstrates that coastal regions with higher population densities are, on average, subsiding (Fig. 1). Local subsidence hotspots are East/Southeast Asian cities like Jakarta (-13.7 mm/year), Tianjin (-13.5 mm/year), Bangkok (-8.5 mm/year), or African cities such as Lagos (-6.7 mm/year) and Alexandria (-4 mm/year), which are all directly observed by InSAR (see also SI Fig. S3).

Because this is new information, as the source data (InSAR, Shirzaei et al., 2024) has not been analyzed before.

L157: Deltas host at least half of the global low elevation coastal population LECZ, (McGranahan et al., 2007), i.e., at least 440 million people, in particular in Asian Megadeltas (e.g., Becker et al., 2024) and the Nile delta.

not a result, a prior knowledge

Agreed, we removed that.

L159: The InSAR estimates [Ohenhen et al., 2025] and dedicated external data-sources of subsidence [Nienhuis et al., 2020, see also SI Fig. S5 for an overview] currently cover coastlines that contain a population of 389 million people, i.e., almost 90% of the global delta LECZ, which presents a substantial improvement in terms of

not a result

We argue that this is a new statistic, since this is not exclusively referring to Ohenhen et al., 2026, and relies on a new assembly of data.

We added: The InSAR estimates [Ohenhen et al., 2026] and dedicated external data-sources of subsidence, **including estimates based on RSETs and GNSS**, [Nienhuis et al., 2020, see also SI Fig. S5 for an overview] currently cover coastlines that contain a population of 389 million people, i.e., almost 90% of the global delta LECZ (**according to**

the DIVA estimates), which presents a substantial improvement in terms of coverage and consistency, especially for the Southeast-Asian deltas [Becker et al., 2024].

L173: Although natural processes such as GIA can explain parts of regional subsidence (up to 2 mm/year), especially along the eastern USA and the western European coastlines [Engelhart and Horton, 2012, Serpelloni et al., 2013, Karegar et al., 2016, Love et al., 2016, Ohenhen et al., 2024], it is widely established that – among other local geological settings and processes - human-induced changes are the main driver of subsidence in most highly urbanized coastal regions. As an example, subsurface fluid extraction was reported to significantly contribute to subsidence in the considered Chinese cities [Ao et al., 2024], Jakarta, Tokyo, Osaka, Manila, Bangkok [Raucoules et al., 2013, Cao et al., 2021], deltas [Minderhoud et al., 2017, Ohenhen et al., 2025], or extended coastlines along the north-eastern US coast [Miller et al., 2013, Johnson et al., 2017]. These processes can occur in parallel with other subsidence processes like natural compaction in deltas [Törnqvist et al. 2008], compaction due to loading of buildings [Ao et al., 2024] although this cause is speculative, or consolidation due to drainage in coastal lowlands such as in New Orleans [Dixon et al., 2006]. Subsidence can additionally be highly non-linear [e.g., Cao et al., 2021, Shirzaei et al., 2021, Törnqvist and Blum, 2024], for example, due to both rapid acceleration in subsidence caused by excessive groundwater extraction and rapid cessation of subsidence when excessive groundwater extraction is stopped, as in Tokyo and Osaka in the 1960s/70s (Fig. 2 in Cao et al., 2021).

not a result but a priori knowledge, plus already largely covered in the Intro

We agree, we removed the entire paragraph.

L174: "western European coastlines" -- which part of western Europe? The max GIA rate in the Netherlands is 0.1 mm/yr for instance, Belgium it is even less. 2 mm/yr is an order of magnitude too high - pls check your source and how they derived it

Yes, we fully agree here. We referred here to the US-East coast, but this paragraph is now removed anyways.

L193: Non-linearities not only complicate extrapolations of the VLM signals, but they can also cause uncertainties in the global estimates due to inconsistent observation periods between measurements (InSAR, GNSS, tide gauges). VLM uncertainties are also influenced by technique-dependent noise, differences in observation-window length, cross-validation uncertainties (e.g., potential offsets between InSAR and GNSS rates), spatial variability and aggregation effects, and parameter uncertainty. We integrate these different kinds of uncertainty in Figure 1d (see Methods, and SI Figs. S6d and S7).

not a result, but a discussion

(see next comment)

L201: It should also be noted, that there is uncertainty regarding the extent to which InSAR-derived subsidence rates may be underestimated, either because shallow subsidence is only partially observed (when reflectors such as buildings are founded on deeper layers), or due to the influence of vertical accretion in dynamic, non-urban landscapes [Törnqvist and Blum, 2024, Minderhoud et al., 2025]. Estimates of these components are usually not available on a global scale, except for the Mississippi delta, where a dedicated subsidence map based on RSETs (Rod-surface elevation tables) is available [Nienhuis et al., 2020] that represents both shallow and deep subsidence. Therefore, partially observed shallow subsidence and accretion are remaining sources of uncertainties in the InSAR derived 209 estimates (see also discussion).

not a result, but a discussion

We fully agree. First, we removed the entire paragraph, as suggested. Second, we moved some of the limitations (interpretation of InSAR) to the beginning of the section, where we also clarify that why we use the RSET estimate (and mention that these issues are later discussed in more detail). Third, we changed the paragraph, such that it's now a more quantitative summary of uncertainties and a short explanation of causes L204:

While densely populated coastlines often experience the highest subsidence rates, we also find that these regions are also associated with the largest uncertainties. VLM uncertainties can be influenced by non-linearities, technique-dependent noise, differences in observation-window length, cross-validation uncertainties (e.g., potential offsets between InSAR and GNSS rates), spatial variability and aggregation effects, and parameter uncertainty. Here, we integrate formal, spatial, and cross-validation uncertainties for InSAR VLM data (based the comparisons with GNSS trends, SI Table S1), as shown in Figure 1d and explained in more detail in the **Methods**, SI Figs. S6d and S7. As a result, median VLM uncertainties are generally higher in the most populated cities (2.6 mm/year) and deltas (1.8 mm/year) compared to all other regions (0.9 mm/year), which are largely controlled by the InSAR uncertainties (SI Fig. S6d). Locally, VLM uncertainties can reach 7–10 mm/year, particularly in some of the most densely populated cities, such as Jakarta and Tianjin (see SI Fig. S3).

L214-216: To understand how subsidence enhances coastal RSL rise, we combine the hybrid VLM estimates with the ASL change (using gridded altimetry data from the Copernicus Marine Service, see Materials and Methods) and compute the contemporary RSL change rates (Fig. 2A).

This is where it becomes interesting, the previous section doesn't add anything new in my opinion.

L223: "global population-weighted mean estimates" -- This is a bit vague - could you somewhere elaborate why this is important? Why express it like this? Does it matter at all? Besides this, I don't think it is known how many people are exactly residing in all those big cities

This refers to one of your major comments (please refer to our more detailed answer). As described, we added a better explanation of why we are doing this: L228: **This distinction is important because coastal populations are highly unevenly distributed, with large numbers of people concentrated in low-lying urban and deltaic regions where RSL change can differ substantially from the coastal-length mean. Population weighting therefore provides a more appropriate measure for assessing human exposure to RSL change and the associated contribution of subsidence on a global scale.**

We also agree that there is uncertainty in estimates of population distribution in the impact assessment models. In response to the previous reviewer (PM) we had thus added: 'It should be emphasized that this analysis isolates only contributions of observational VLM uncertainties from other uncertainty sources present in digital elevation models, protection estimates, and population data, which are fundamental for coastal risk assessments (Methods).

L224-225: opposed to what the average coastal area experiences," -- I think this is way more important for policy, engineering, etc.

We added L233: **By contrast, area-weighted estimates may be more relevant for applications focused on coastal land changes.**

L240: "frequent in regions with higher population." -- it strongly depends on how the subsurface is used and what the subsurface properties are. Tokyo, Bangkok, Shanghai are prime examples of Asian megacities that stalled subsidence by changing groundwater policies. Most major European cities do not extract water from underneath the cities anymore since the 1980s.

We fully agree, later we discuss that although this is a global tendency there is strong local variation (L313: However, it should be emphasized that this relationship is strongly skewed and large coastal cities (...) and densely populated deltas (...) are affected by strong subsidence and thus increased RSL change.).

L251: results by NI21b (dotted line, representing the average between their upper and lower estimates) substantially

i said it before, it is difficult to assess statements like this as a reader if you do not know how the NI21b analysis was set-up

Please refer to the associated major comment above.

L245: "uplift, will" -- see previous comment, most cities that stopped groundwater pumping are actually uplifting because pore pressures are naturally recovering in aquifer systems

We changed this to:

In contrast, areas that are uplifting by at least 1 mm/year contain less than 10% of the LECZ population, indicating that processes **such as** GIA, or tectonic uplift, **as well as managed groundwater recharge [e.g., Shirzaei et al., 2021], will-currently** only attenuate RSL rise for a small fraction of the LECZ population.

L267: Therefore, subsidence (with a global population-weighted average of 2.8 mm/year) currently contributes almost as much to the population-weighted RSL change as the climate-driven ASL component (3.15 mm/year).

i cannot follow this, you should realize that you conducted on paper a very nice analysis (combining global ASL with VLM), but then you are blurring it with population statements making it very difficult to follow

We hope this became clear now with our previous changes/comments: population-weighting is very important to understand potential impacts of subsidence on coastal risk to people.

L303: "Bangkok" -- i thought subsidence largely ceased in Bangkok due to groundwater policy changes in the 1990s

We are thankful for this comment, as this directly addresses the problem of the high spatial variability in the cities: Subsidence is still very high in some places in Bangkok and associated with high spatial variability (please refer to SI Fig. S3), also showing uplift in some areas. We clarified that we refer here to the estimates based on the VLM data on the coastal grid, and that variability within some cities can be extremely large. Local subsidence hotspots are East/Southeast Asian cities like Jakarta (-13.7 mm/year), Tianjin (-13.5 mm/year), Bangkok (-8.5 mm/year), or African cities such as Lagos (-6.7 mm/year) and Alexandria (-4 mm/year), ~~which are all directly observed by InSAR (see also SI Fig. S3).~~ These estimates are derived from InSAR observations [see also SI Fig. S3] and represent coastal averages of the aggregated VLM data on the DIVA grid. However, subsidence rates can vary substantially within some of the fastest-subsiding cities. In Jakarta, for example, some areas subside at rates of up to -42 mm/year, while more central parts experience uplift of up to +15 mm/year [0.1th and 99.9th percentiles; see also Extended Data File 2]. Similar, though generally less pronounced, spatial contrasts are also found in other cities such as Bangkok and Ho Chi Minh City. As a result, aggregated coastal city-scale estimates remain associated with considerable uncertainty, and local risk assessments require careful consideration of high-resolution spatial variability, as well as the specific infrastructure and populations affected.

That subsidence is an ongoing issue after the 2000s is also shown, for example, here: https://tice.buu.ac.th/knowledge/IJRS_2013.pdf

L314: The data indicates that these highly populated regions are more likely to coincide with higher-than-average subsidence, which explains why humans experience RSL changes that are significantly higher than the spatially averaged rates.

This is way too simple, human-induced subsidence is driven by policy, not by the fact that many people live in an area. Drainage policy, construction policy, soil settlement policy, extraction policy etc. You are too much focused on an empirical relation without understanding what actually is happening.

Thank you for this suggestion. We clarified that this is a purely statistical global-scale relationship and added your suggestions: L325: The data indicates that these highly populated regions are **statistically** more likely to coincide with higher-than-average subsidence **on a global scale**, which explains why humans experience RSL changes that are significantly higher than the spatially averaged rates. **However, understanding the causes of these statistical relationships requires consideration of interacting physical processes and human management decisions, including drainage policy, construction practices, soil settlement, resource extraction, and broader subsurface-use governance.**

L322-324: The contemporary population-density weighted global-mean RSL change of 6 mm/year is nearly three times the coastal-length weighted global-mean RSL change (2.1 mm/year).

what does this physically means?

These statistics indicate the average RSL change that is currently experienced by coastal populations, and highlights the usefulness to consider population distribution, as area- or coastal length-weighted averages may completely miss the relevance of subsidence in causing this effects.

L329-330: subsidence is a significant coastal hazard and contributor to risk.

the first sea level rise enhancing subsidence maps in the Netherlands stem from 1918. In other words, you cannot claim this as new, it is a priori knowledge. In fact, it is probably the reason you conducted this study in the first place

We deleted this statement.

L337-L339: NI21b, which were mostly derived from the literature and expert judgement rather than geodetic measurements, or by missing observations in the applied VLM observation data due to their partially sparse spatial distribution.

this is the first clue in the manuscript how NI21b derived its results. If it is based on expert judgment, why do you keep comparing your observations to it? Why

the obsession? What you did is way more important. I think this section on comparison should be removed, it doesn't make sense to compare observations to opinions

We thank the reviewer for this comment, and refer to the associated major comment, where we explain why we compare our results to NI21b. We added some more information in the comment below.

L342-346: In contrast, NI21b considered 138 large coastal cities (more than one million people in 2005) and 113 deltas worldwide. However, they lacked information on most Chinese coastal cities which are available to this analysis. Other estimates (e.g., Syvitski et al. 2022) provide information on 89 large deltas (>1000km²), 885 medium deltas (1000 to 10km²) and about 1460 small deltas (<10km²) worldwide.

i am now confused, so NI21b did use observations of all these areas? How can anyone know how much subsidence there is at any location without monitoring? Again, in general, if you want to keep comparing this study to NI21b, you should crystal clear explain what they did and what they didn't do, and why that matters and why you think it is relevant to compare yourself (i don't see the relevance btw)

We improved this further and clarified the number of expert judgements: L356: However, they **relied on expert judgments in 77 deltas and 8 cities, and** lacked information on most Chinese coastal cities which are available to this analysis. ~~Other estimates (e.g., Syvitski et al. 2022) provide information on 89 large deltas (>1000 km²), 885 medium deltas (1000 to 10 km²) and about 1460 small deltas (<10 km²) worldwide.~~

Please also refer to our previous comment on the relevance.

L349 This indicates that missing data still plays a role, despite the substantial progress in covering the largest subsiding cities and deltas

this doesn't matter. anyone working in the field of vlm/subsidence knows this is always the case, you don't have to mention this

We deleted this.

L359: to what extent they are influenced by vertical accretion processes.

this should be removed. Insar cannot be used for sedimentation monitoring

We fully agree. Note, that we did not mean to imply that it can be used for this purpose. We rather state here that there is uncertainty about how vertical accretion processes could potentially influence VLM rates. We added this caveat as a response to the last reviewer, who noted: *The InSAR used (depending on type and processing of the InSAR) may/may not include vertical accretion to some extent – although to which*

extent remains uncertain for such large scale studies with limited validation on their effectiveness to register accumulation under different circumstances, environments and vegetation types.

This is also discussed in Törnqvist and Blum, 2024, MD <https://pmc.ncbi.nlm.nih.gov/articles/PMC12337573/> , which we added now as well.

L363: "who also aggregated data based on such in situ information." -- again a little piece of the puzzle, please lump the info at the beginning

This is now in the introduction.

L363-366: reflectors such as buildings have different foundation depths, this can lead to high spatial variability due to differential settlement between these reflectors. Buildings with deeper foundations are also less affected by shallow subsidence.

and therefore you should filter them out

We thank the reviewer for this comment. Our intention was to emphasize that InSAR measurements obtained from built structures may reflect localized deformation behavior that depends on structural and geotechnical characteristics. In particular, differences in foundation depth can produce differential settlement signals between reflectors, contributing to the spatial variability observed in InSAR-derived vertical land motion fields. Filtering out reflectors associated with deep foundations would require prior knowledge of the foundation depth of each reflector across global coastal zones. Such information is almost never available at the spatial scales covered by our dataset, making systematic filtering operationally infeasible in a global analysis. Moreover, the differential motion between shallow- and deep-founded structures does not represent noise but rather a real physical signal that reflects the heterogeneity of subsidence processes and the differential response of the built environment. Removing these reflectors would therefore risk masking important spatial variability rather than resolving it. Preferentially retaining only shallow-foundation reflectors would also introduce a selection bias in densely urbanized coastal regions, where large engineered structures with deeper foundations often dominate the coherent scatterer population and are highly relevant for coastal risk assessments. Importantly, this limitation is explicitly acknowledged in the manuscript (L374–381), where we discuss the potential for deep-founded structures to underestimate shallow subsidence signals. We further account for this heterogeneity in the uncertainty framework through the spatial uncertainty component (σ_{spatial}), which captures the variability of vertical land motion rates aggregated across reflectors with differing foundation conditions within each coastal segment. We also added the following: **Buildings with shallow foundations respond mainly to near-surface compaction, while pile-supported structures are coupled to deeper, more stable layers and primarily experience deformation occurring at depth. Because InSAR measures displacement of surface scatterers on buildings, the signal may exaggerate risk for structures founded shallowly while underrepresenting stresses affecting deep-founded buildings. This depth-dependent decoupling introduces**

uncertainty in hazard interpretation, particularly in cities with mixed foundation systems, and highlights the need to interpret InSAR observations together with subsurface stratigraphy and foundation information.

L372: "continuous global GNSS networks (e." -- i dont agree, a gnss station has a very limited lateral reach (VLM rates changes in lateral direction sub-meter scale), plus you need to know the foundation level of each station, plus you need to know which subsurface processes are influencing the station. If you want to make a proper VLM map of global coastal zones, i would put my money on good processed InSAR

We fully agree with the limitations of GNSS in terms of spatial resolution (i.e., they are point measurements) and issues with the (unknown) foundation levels and neglect of shallow subsidence. We thus mentioned in line: 'Comparable to limitations in InSAR estimates, GNSS reference stations with deep anchor depths may overlook shallow compaction in landscapes where such processes are significant'. Despite these limitations, we highlighted the importance of these networks (also as a response to the previous reviewer 2) because GNSS 'measurements are the prerequisite to derive comparable InSAR VLM estimates in a global reference frame and are thus fundamental to reducing biases in regional estimates.' Hence they are needed to obtain observations in a consistent reference frame.

We appreciate the reviewer's comment and agree that the role of GNSS networks should be described more carefully. GNSS stations provide highly accurate point measurements of vertical land motion and are incredibly useful to establish an absolute geodetic reference frame for InSAR time series. However, their spatial sampling is sparse and they cannot capture the strong lateral variability in vertical land motion that may occur over short spatial scales in coastal and urban environments. In contrast, InSAR observations provide dense spatial coverage and are therefore better suited for resolving fine-scale deformation patterns. The original framing therefore overstated the role of GNSS networks in resolving spatial VLM variability, and we have revised the manuscript to clarify that the role of GNSS here is specifically as a geodetic reference frame anchor for InSAR processing, emphasizing the complementary nature of these two datasets rather than implying GNSS as a standalone mapping tool.

We mention these limitations in L397: 'Comparable to limitations in InSAR estimates, GNSS reference stations with deep anchor depths may overlook shallow compaction in landscapes where such processes are significant'. We also added: L399: **Another limitation is that GNSS stations only provide point-measurements and, unlike InSAR, cannot fine-scale deformation patterns.** Therefore, given the key role of deltas and cities in controlling the rates-of sea-level rise felt by the average person, these factors must be considered, and InSAR data should ideally be analyzed in synergy with in-situ (e.g., **RSETs-extensometers** and GNSS) observations.

L383: "RSETs" -- RSETs require people to go into the field, it will never work. I would change it with extensometers, they have automated loggers that are connected to servers

We changed that accordingly.

L463: in Europe (EGMS) (see also Thiéblemont et al., 2024), USA [Ohenhen et al., 2024], New Zealand [Hamling et al., 2022], 26 Chinese coastal cities [Ao et al., 2024], 15 other selected coastal cities around the world [Shirzaei et al., 2024], and 40 deltas

see my previous remark on time windows of the different data sets, mention these and if they don't overlap, explain how you deal with this during the interpretation

Thank you for that comment. We added that in the beginning of the first section (describing the global dataset). 'Our second assumption is that trends from all data sources are representative of the changes over the entire period (1995-2021) considered here. This is a simplification, as InSAR based VLM rates are generally derived from relatively short and variable observation periods (~5-15 years, see SI Table S1). Uncertainties arising from potentially unobserved non-linear changes, as well as from partially observed shallow subsidence and vertical accretion, are therefore discussed further in the Discussion.'

L517: "Rhine," -- the Rhine doesn't form a delta btw, it doesn't have enough sediment for that. Hence the Netherlands has a lot of peat, hence the area is subjected to subsidence. It is a coastal plain

We added 'deltas and coastal plains'

L987: "Figure 1" -- how far land inwards does the ribbon stretches? is it a fix width along the coast or did you use some LECZ maps to clip the Insar?

The data is visualized on the coastal DIVA grid points (i.e., a coastline), that are representative of coastal segments of the LECZ. We added: 'on the DIVA coastal segments.'

*** Page #41:**

> Hybrid estimate of vertical land motion (VLM) along the global coastlines

this is the most important aspect of this study. Could you please provide some zoom-ins as well?

Thank you for this idea. We updated it as shown:

Caption: **Figure 1. Hybrid estimate of vertical land motion (VLM) along the global coastlines.** The main map (a) shows uplift (positive signals) and subsidence in mm/year on the DIVA coastal segments. Values outlined by black circles present regions where InSAR data is available, or where single GNSS estimates are used. The points are scaled by their absolute value to highlight strong subsidence or uplift. Panels b–f show individual InSAR estimates for: (b) the United States [Ohenhen et al., 2024], (c) Europe from EGMS (see also Thieblemont et al., 2024), (d) the Nile Delta [Ohenhen et al., 2026], (e) Lagos [Shirzaei et al., 2024], and (f) New Zealand [Hamling et al., 2022]. For Europe and the United States, we show the coastal low-resolution DIVA grid points (12,148 elements), whereas for the other regions we show the high-resolution grid (247,666 grid points; see also **Methods**). (bg) and (eh) show the fraction of coastal population (length) covered by the individual datasets (InSAR, GIA, GPS, and the interpolated data from OE24). The colors depict the population- (or length-) weighted vertical land motion (in mm/year) of the individual datasets. (di) displays the estimated 1σ VLM uncertainties of the datasets (using the same color scale as in (a)), which are computed from the formal, spatial and cross-validation uncertainties in the InSAR data, and the provided uncertainties from OE24 and Caron et al., 2024 (see **Methods**).

a

b

c Europe (EGMS, see also Thiéblemont et al., 2024)

d

e

f

g

h

i

*** Page #42: "Cont" -- please provide some zoom-in panels**

We updated this as follows (the VLM map above already gives the most important details). Note that all plots will be provided at much higher resolution as vector graphics.

Figure 2. Contribution of VLM to relative sea-level change. a) RSL change [mm/year] as the combination of ASL change (from CMEMS over 1995-2019) and VLM (based on the VLM reconstruction OE24), GIA [Caron et al. 2018], where OE24 has missing data) and InSAR from EGMS, Ohenhen et al., 2024, 2025, Shirzaei et al., 2024, and Ao et al., 2024. **b)** We also shows the number of people living below 10 m elevation on a logarithmic scale (black colorbar). For illustrative purposes, we applied radial basis function smoothing with a 120 km length scale to the coastal population data at the DIVA segment grid points to emphasize global population hotspots. The size of the data points are non-linearly scaled by their value. **cb)** Cumulative distribution of the contribution of coastal length-weighted and population-weighted VLM to RSLC (positive sign = subsidence). The inverted cumulative frequency on the y-axis thus refers here to the share of population experiencing at least the subsidence rate shown on the x-axis. Solid lines represent the hybrid VLM estimates of this study, the dashed line represents the same dataset without InSAR, and the dotted line represents the total averaged VLM estimate from NI21b. The shaded region surrounding the solid red line (i.e., the VLM estimate) denotes the 95% confidence interval.

This uncertainty is estimated from a bootstrapped distribution of CDFs generated by perturbing the VLM rates using normally distributed random errors derived from the estimated uncertainties (see Methods). We also show the fraction of people who experience subsidence rates of >0 , >1 , and >2 mm/year (using estimates from the hybrid reconstruction). **dc** shows both, the averaged, population-weighted RSL change per country (as indicated by the colors ranging from blue to red, i.e., low to high RSL change), as well as the total coastal population, which is displayed by the modulation in the transparency of the colors.

Good luck, Kay Koster